# Decomposing the Basic Abilities of Large Language Models: Mitigating Cross-Task Interference in Multi-Task Instruct-Tuning

Bing Wang [1 2]  Ximing Li [1 2 3]  Changchun Li [1 2]  Jinjin Chi [1 2]  Gang Niu [3]  Masashi Sugiyama [3 4]

## Abstract

Recently, the prominent performance of large language models (LLMs) has been largely driven by multi-task instruct-tuning. Unfortunately, this training paradigm suffers from a key issue, named cross-task interference, due to conflicting gradients over shared parameters among different tasks. Some previous methods mitigate this issue by isolating task-specific parameters, e.g., task-specific neuron selection and mixture-of-experts. In this paper, we empirically reveal that the cross-task interference still exists for the existing solutions because of many parameters also shared by different tasks, and accordingly, we propose a novel solution, namely Basic Abilities Decomposition for multi-task Instruct-Tuning (BADIT). Specifically, we empirically find that certain parameters are consistently co-activated, and that co-activated parameters naturally organize into base groups. This motivates us to analogize that LLMs encode several orthogonal basic abilities, and that any task can be represented as a linear combination of these abilities. Accordingly, we propose BADIT that decomposes LLM parameters into orthogonal high-singular-value LoRA experts representing basic abilities, and dynamically enforces their orthogonality during training via spherical clustering of rank-1 components. We conduct extensive experiments on the *SuperNI* benchmark with 6 LLMs, and empirical results demonstrate that BADIT can outperform SOTA methods and mitigate the degree of cross-task interference.

[1]College of Computer Science and Technology, Jilin University [2]Key Laboratory of Symbolic Computation and Knowledge Engineering, Ministry of Education, Jilin University [3]RIKEN Center for Advanced Intelligence Project [4]Graduate School of Frontier Sciences, University of Tokyo. Correspondence to: Ximing Li <liximing86@gmail.com>.

*Proceedings of the 43rd International Conference on Machine Learning*, Seoul, South Korea. PMLR 306, 2026. Copyright 2026 by the author(s).

## 1. Introduction

Recently, the community has witnessed the rapid development of *large language models* (LLMs) in tackling a variety of downstream tasks, e.g., reasoning (Yang et al., 2025a) and Q&A (Hu et al., 2025). This remarkable capability is primarily attributed to the supervised fine-tuning across massive data of diverse tasks, namely *multi-task instruct-tuning* (Wang et al., 2023b; Dou et al., 2024; Shi et al., 2026).

Despite the success of LLMs, their multi-task instruct-tuning faces a key challenge referred to as *cross-task interference*, that is, training multiple tasks simultaneously inevitably results in conflicting gradients from different task data over shared LLM parameters, thereby degrading performance on each task (McCloskey & Cohen, 1989; Luo et al., 2023). To handle this, a basic assumption in multi-task instruct-tuning is that *different tasks consistently activate distinct subsets of LLM parameters* (Leng & Xiong, 2025; Tang et al., 2025). Accordingly, some previous methods identify and isolate task-specific parameters (Tang et al., 2021; Zhao et al., 2024b). For example, Leng & Xiong (2025) locates task-specific neurons in LLMs through gradient attribution and selectively trains only those parameters. Another line of research applies *mixture-of-experts* (MoE) to separate the parameters associated with different tasks (Wang et al., 2023a; Dou et al., 2024), incrementally introducing multiple experts to explicitly isolate task-specific parameters, while enforcing orthogonality constraints among the parameters of different experts (Wang et al., 2023a).

In this paper, we empirically reveal that these existing solutions still suffer from cross-task interference because of many parameters still shared by different tasks, and accordingly, we propose a novel solution to further mitigate it. Specifically, we perform empirical evaluations with standard methods on the benchmark dataset with 15 different tasks. The experimental results in Figs. 1 and 2 reveal the significant overlap among the task-specific parameters cross different tasks. We analyze 15 tasks (Wang et al., 2022b), and illustrate how they activate different neurons, parameters, and MoE experts across various LLMs, respectively. The experimental results show that the majority of neurons, parameters, and MoE experts are simultaneously activated by multiple tasks. Therefore, prior multi-task instruct-tuning

methods fail to fully isolate task-specific parameters, leaving a substantial number of shared parameters that still exhibit conflicting gradients across tasks during optimization.

To further mitigate cross-task interference, we propose a new method, namely *Basic Abilities Decomposition for multi-task Instruct-Tuning* (BADIT). Specifically, BADIT is inspired by an empirical observation in Fig. 1 that certain neurons and parameters are always co-activated across tasks, and these co-activated ones can be naturally grouped into several base groups. This motivates us to *decompose LLM parameters into a set of orthogonal base groups, named **basic ability**, and formulate each ability as an MoE expert*, thereby mitigating cross-task interference by isolating these abilities, instead of tasks. To implement this idea, BADIT involves two key strategies. First, we decompose the LLM parameters via singular value decomposition (Meng et al., 2024) into a linear combination of multiple large-singular-value *low-rank adaptation* (LoRA) experts (Hu et al., 2022) and a residual term comprising low-singular-value ones. Since the decomposed LoRA experts are naturally orthogonal, they serve as initial basic abilities. Second, during multi-task instruct-tuning, to ensure that each LoRA expert consistently represents a distinct basic ability, we introduce a dynamic grouping strategy based on spherical clustering. It enforces orthogonality among the optimization gradients of different LoRA experts while maintaining similar angles among the rank-1 components within each LoRA.

To evaluate the performance of BADIT, we conduct experiments on 6 different LLMs using *SuperNI* (Wang et al., 2022b), a multi-task dataset encompassing 15 diverse tasks. Generally, BADIT outperforms the state-of-the-art multi-task instruct-tuning approach, GainLoRA (Liang et al., 2025), by an average of 2.68 ROUGE scores, and quantitatively demonstrates reduced cross-task interference. *Our source code has been released in the repository* `https://github.com/wangbing1416/BADIT`.

Our contributions can be summarized as three-fold:

- Our experiments reveal that different tasks activate shared parameter subsets, and certain shared ones are always co-activated across tasks. We analogize the co-activated subsets as the LLM's basic abilities.

- We propose BADIT, which decomposes the basic abilities from the LLM's parameters as MoE experts and represents any task as a combination of these abilities.

- Extensive experiments are conducted to demonstrate that BADIT can alleviate cross-task interference.

## 2. Preliminary Experiments

In this section, we present preliminary experiments demonstrating that, during the multi-task instruct-tuning of LLMs,

their activated parameters can be organized as multiple base groups (we refer to them as *basic abilities*).

### 2.1. Experimental Settings

Following Zhao et al. (2024b) and Liang et al. (2025), we conduct an empirical study using 15 different tasks from *SuperNI* (Wang et al., 2022b), a multi-task instruct-tuning dataset. Our investigation analyzes activation patterns of neurons and parameters in Llama3-3B (AI@Meta, 2024) and Qwen3-4B (Yang et al., 2025a), and evaluates their task-shared activations. Furthermore, we integrate a lightweight MoE architecture, termed LoRAMoE (Dou et al., 2024), into these LLMs to examine how different tasks activate these experts when each task is trained independently. We fix the number of LoRA experts at 8 and select the top-4 LoRA experts with the highest routing weights.

**How to identify activated neurons and parameters across tasks?** To investigate how different task datasets influence LLM parameters, we begin by analyzing the shared **neurons** activated by these tasks. Formally, given a dataset $\mathcal{D}_t$ for the $t$-th task, an activated neuron is defined as one hidden state whose output in a forward pass is non-zero or significantly different from zero. For example, consider the average activation in a feed-forward layer computed over the task dataset $\mathbf{h} = \frac{1}{|\mathcal{D}_t|} \sum_{\mathbf{x} \in \mathcal{D}_t} \sigma(\mathbf{W}\mathbf{x} + \mathbf{b})$, a neuron $h_i$ is deemed activated if $|h_i| > \epsilon$, with $\epsilon$ being a small positive threshold. Additionally, following Song et al. (2024), we also examine activated LLM parameters through **gradients** and assess the task-shared parameters. Specifically, given $\mathcal{D}_t$, we calculate the Fisher information matrix (Wu et al., 2024) towards the gradient of a specific parameter matrix $\mathbf{W}_i$ in full LLM parameters $\boldsymbol{\theta}$. We then use the diagonal entries of this matrix, each representing the expected variance of the score function for the corresponding parameter, as a measure of parameter activation as follows:

$$F_{ii} = \mathbb{E}_{\mathbf{x} \sim \mathcal{D}_t} \left[ \left( \frac{\partial}{\partial \mathbf{W}_i} \log p_{\boldsymbol{\theta}}(y|\mathbf{x}) \right)^2 \right], \ \mathbf{W}_i \in \boldsymbol{\theta},$$

where we implement $\log p_{\boldsymbol{\theta}}(y|\mathbf{x})$ by the supervised fine-tuning loss. Prior work suggests that LLMs primarily store knowledge in their feed-forward networks (Geva et al., 2021; Dai et al., 2022), particularly within their gating layers (Song et al., 2024). Consequently, our subsequent experiments examine the gating layer, and the results for other components, e.g., self-attention layers, and additional analyses across more LLMs, are provided in Appendix H.1.

### 2.2. Task-Specific Activations Are Still Shared

The activated neurons and parameters of the two LLMs are visualized in Fig. 1. The two left subplots depict neuron activation patterns, where each row corresponds to the set of neurons activated for a specific task. The two right

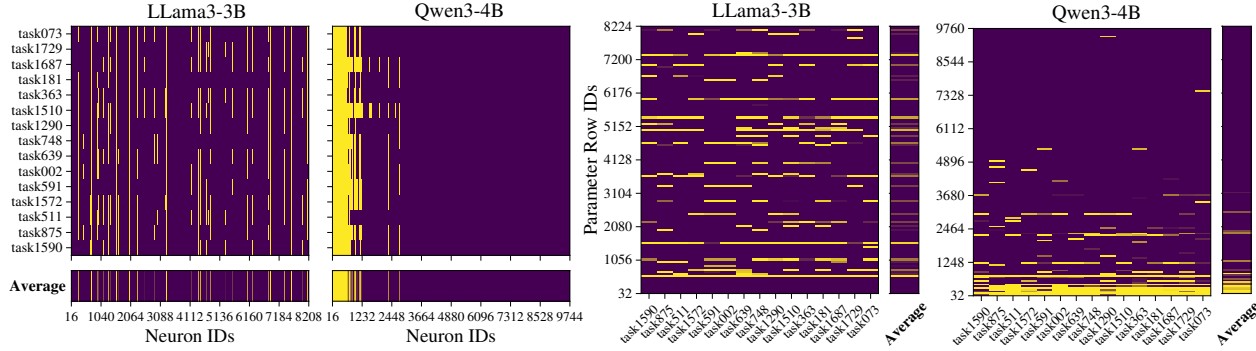

*Figure 1.* Shared task-specific neurons and parameters across different tasks.

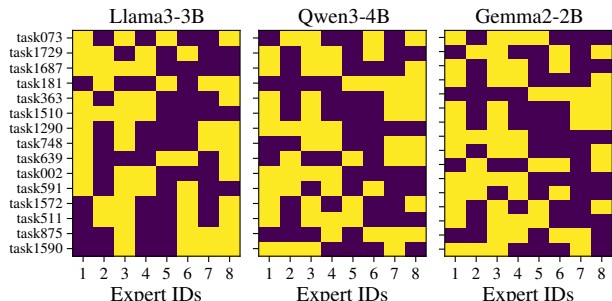

*Figure 2.* Activated experts in MoE-based LLMs by different tasks.

subplots display parameter-level activations, where each column represents a specific task and indicates the number of activated parameters in the corresponding row of the parameter matrix for that task. Additionally, Fig. 2 illustrates how different tasks activate the 8 MoE experts, based on the routing weights produced by the gate module in the final layer of the LLMs. In all figures described above, activated components are marked in yellow.

The experimental results in Fig. 1 show that, *for each task, most activated neurons or parameters are shared with those of other tasks*. Some neurons and parameters are even shared across 15 tasks simultaneously. Notably, in Qwen3-4B, the activated neurons and parameters are predominantly shared among 15 tasks at lower IDs. The results in Fig. 2 also reveal that even with a relatively advanced MoE architecture, a substantial number of experts are jointly activated across tasks. These shared experts still suffer from interference due to conflicting gradients arising from different tasks.

### 2.3. LLMs Can be Decomposed as Basic Abilities

Based on the results presented in Fig. 1, we can draw several preliminary observations. First, whether considering neurons or LLM parameters, activations are consistently concentrated within a small subset, while the vast majority remain inactive. Second, we further observe that certain neurons and parameters are consistently co-activated across multiple tasks. These co-activated units tend to form a small

number of recurring base activation patterns. For example, in both Llama3-3B and Qwen3-4B, there exist specific neurons and parameters that are simultaneously activated by exactly 15 tasks. These observations suggest that such co-activated neurons and parameters likely fulfill shared functional roles—akin to the model possessing a repertoire of basic, reusable abilities, with each complex task being realized through a composition of these underlying primitives. This insight motivates our core approach: *decomposing the distinct, orthogonal basic abilities* inherently encoded in LLMs. By isolating these basic units, we aim to disentangle task-specific interference.

## 3. The Proposed BADIT Method

In this section, we first present the technical preliminaries in Sec. 3.1. We then provide an overview of our proposed BADIT method in Sec. 3.2. The two core steps of BADIT are described in detail in Secs. 3.3 and 3.4, respectively.

### 3.1. Technical Preliminaries

**Multi-task instruct-tuning.** Given $T$ natural language tasks $\{\mathcal{D}_t\}_{t=1}^T$, $\bigcap_{t=1}^T \mathcal{D}_t = \varnothing$, where $\mathcal{D}_t = \{(\mathbf{x}_i^t, y_i^t)\}_{i=1}^{|\mathcal{D}_t|}$ represents task-specific instruct samples, the goal of multi-task instruct-tuning is to tune a pre-trained LLM $\mathcal{F}(\cdot\,;\boldsymbol{\theta}^0)$ across tasks while mitigating cross-task interference. Formally, for the task $\mathcal{D}_t$, we optimize LLM parameters as

$$\boldsymbol{\theta}^t = \arg\min_{\boldsymbol{\theta}} \frac{1}{|\mathcal{D}_t|} \sum_{(\mathbf{x}_i^t, y_i^t)\in\mathcal{D}_t} \left[ \mathcal{L}\big(\mathcal{F}(\mathbf{x}_i^t;\boldsymbol{\theta}), y_i^t\big)\right], \quad (1)$$

where $\mathcal{L}(\cdot,\cdot)$ is a supervised fine-tuning loss.

**Low-rank adaptation (LoRA).** LoRA is a parameter-efficient fine-tuning method that injects trainable low-rank matrices into each LLM weight matrix. During fine-tuning, the LLM weight matrix $\mathbf{W} \in \mathbb{R}^{m\times n}$ (e.g., MLP layers) remains frozen, and the adapted forward pass becomes

$$\mathbf{h} = (\mathbf{W} + \Delta\mathbf{W})\mathbf{x} = (\mathbf{W} + \mathbf{AB})\mathbf{x}, \quad (2)$$

where $\mathbf{A} \in \mathbb{R}^{m\times r}$ and $\mathbf{B} \in \mathbb{R}^{r\times n}$ are trainable low-rank matrices, $r \ll \min(m,n)$ controls the rank w.r.t. parameter

efficiency. Additionally, $\mathbf{A}$ and $\mathbf{B}$ are respectively initialized by the *Kaiming* initialization (He et al., 2015) and a zero matrix to keep the initial $\Delta\mathbf{W}\mathbf{x} = \mathbf{0}$, preserving the pre-trained LLM's behavior at the start of fine-tuning.

**Singular value decomposition (SVD).** SVD is a classical matrix factorization algorithm, and any matrix $\mathbf{W} \in \mathbb{R}^{m \times n}$ can be decomposed into a product of matrices as follows:

$$\mathbf{W} = \mathbf{U}\boldsymbol{\Sigma}\mathbf{V}^\top, \qquad (3)$$

where $\mathbf{U} \in \mathbb{R}^{m \times m}$ and $\mathbf{V} \in \mathbb{R}^{n \times n}$ are orthogonal matrices whose columns correspond to the left and right singular vectors of $\mathbf{W}$, respectively, and $\boldsymbol{\Sigma}$ is a diagonal matrix containing the singular values $\sigma_1, \dots, \sigma_{\min(m,n)}$ arranged in descending order. Previous works, e.g., *principal singular values and singular vectors adaptation* (PiSSA) (Meng et al., 2024; Yang et al., 2024), apply SVD in Eq. (3) to the LLM weight matrix $\mathbf{W}$ to identify its principal components, and formulate a LoRA analogous to Eq. (2) as follows:

$$\mathbf{h} = \mathbf{W}\mathbf{x} \xrightarrow{\text{SVD}} \mathbf{U}\boldsymbol{\Sigma}\mathbf{V}^\top\mathbf{x} \xrightarrow{\text{truncate}} (\mathbf{A}\mathbf{B} + \widehat{\mathbf{W}})\mathbf{x}, \quad (4)$$

where $\widehat{\mathbf{W}}$, $\mathbf{A}$, and $\mathbf{B}$ are respectively the residual matrix and two low-rank matrices as follows:

$$\mathbf{A} = \mathbf{U}_{[:r]} \operatorname{diag}\!\big(\sqrt{\boldsymbol{\Sigma}_{[:r]}}\big), \ \mathbf{B} = \operatorname{diag}\!\big(\sqrt{\boldsymbol{\Sigma}_{[:r]}}\big)\mathbf{V}^\top_{[:r]},$$
$$\widehat{\mathbf{W}} = \mathbf{U}_{[r:]} \operatorname{diag}\!\big(\boldsymbol{\Sigma}_{[r:]}\big)\mathbf{V}^\top_{[r:]}. \qquad (5)$$

where $\operatorname{diag}(\cdot)$ denotes the diagonal matrix whose values are the singular values of the matrix.

### 3.2. Method Overview

The primary idea of BADIT is to decompose the basic abilities embedded within the LLM parameters and leverage these abilities to represent any downstream task. Generally, our BADIT method consists of two main steps: *Basic ability decomposition* (BAD) disentangles an initial set of orthogonal LoRA experts from the LLM's parameter matrices, where each expert corresponds to a different basic ability; *Dynamically orthogonal grouping* (DOG) dynamically groups rank-1 components within LoRA during training to preserve orthogonality among these basic abilities. The overall framework of BADIT is depicted in Fig. 3.

Specifically, given a pre-trained LLM $\mathcal{F}(\cdot\,;\boldsymbol{\theta}^0)$, the BAD step begins by decomposing each pre-trained weight matrix $\mathbf{W}^0 \in \boldsymbol{\theta}^0$ into a weighted combination of multiple LoRA components, as $\mathbf{W}^0 = \sum_{k=1}^{K} \alpha_k \mathbf{A}_k \mathbf{B}_k + \widehat{\mathbf{W}}$, using the extended version of the SVD method in Eq. (4), where $K$ represents the number of LoRA experts, $\alpha_k$ is a learnable routing weight, and $\widehat{\mathbf{W}}$ represents the residual term. By the decomposition, we interpret each LoRA component $\mathbf{A}_k \mathbf{B}_k$ as an initial basic ability embedded within the pre-trained weights. During the supervised fine-tuning process

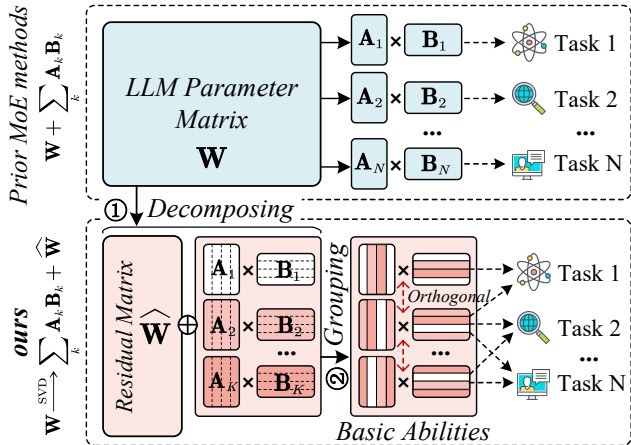

*Figure 3.* Comparison with prior multi-task instruct-tuning methods. We decouple key LLM parameters into basic abilities (i.e., LoRA experts) and dynamically group their rank-1 components, enforcing orthogonality to represent distinct basic abilities.

in Eq. (1), we freeze the residual weights $\widehat{\mathbf{W}}$ and solely update the LoRA parameters.

To further encourage each LoRA expert to specialize in a different basic ability during fine-tuning, our method follows the hypothesis that *the training gradients of different basic abilities should be mutually orthogonal* (Lin et al., 2022a;b). To implement this principle, the DOG step introduces a dynamic adaptation mechanism that progressively groups the rank-1 components of the initialized LoRA experts $\{\mathbf{A}_k \mathbf{B}_k\}_{k=1}^K$ during the fine-tuning of the LLM. This grouping strategy actively enforces orthogonality among the training gradients of different LoRA modules. We provide detailed descriptions of the BAD and DOG steps in Secs. 3.3 and 3.4, respectively.

### 3.3. Basic Ability Decomposition

The BAD step aims to decompose the pre-trained weight matrices $\forall\, \mathbf{W}^0 \in \boldsymbol{\theta}^0$ of the LLM into multiple LoRA experts, each intended to capture a different basic ability implicitly encoded in $\mathbf{W}^0$. Inspired by previous SVD methods on LLM parameters (Meng et al., 2024; Yang et al., 2024), we decompose $\mathbf{W}^0$ as follows:

$$
\begin{aligned}
\mathbf{W}^0 \xrightarrow{\text{SVD}} & \mathbf{U}_{[:rK]}\boldsymbol{\Sigma}_{[:rK]}\mathbf{V}^\top_{[:rK]} + \mathbf{U}_{[rK:]}\boldsymbol{\Sigma}_{[rK:]}\mathbf{V}^\top_{[rK:]} \\
= & \Big(\mathbf{U}_{[:rK]}\operatorname{diag}\!\big(\sqrt{\boldsymbol{\Sigma}_{[:rK]}}\big)\Big)\Big(\operatorname{diag}\!\big(\sqrt{\boldsymbol{\Sigma}_{[:rK]}}\big)\mathbf{V}^\top_{[:rK]}\Big) \\
& + \mathbf{U}_{[rK:]}\boldsymbol{\Sigma}_{[rK:]}\mathbf{V}^\top_{[rK:]} \qquad (6) \\
\xrightarrow{\text{truncate}} & \sum_{k=1}^{K}\Big(\mathbf{U}_{[r(k-1):rk]}\operatorname{diag}\!\big(\sqrt{\boldsymbol{\Sigma}_{[r(k-1):rk]}}\big)\Big) \\
& \Big(\operatorname{diag}\!\big(\sqrt{\boldsymbol{\Sigma}_{[r(k-1):rk]}}\big)\mathbf{V}^\top_{[r(k-1):rk]}\Big) \\
& + \mathbf{U}_{[rK:]}\boldsymbol{\Sigma}_{[rK:]}\mathbf{V}^\top_{[rK:]} = \sum_{k=1}^{K} \mathbf{A}_k \mathbf{B}_k + \widehat{\mathbf{W}},
\end{aligned}
$$

where $\mathbf{A}_k\mathbf{B}_k$ denotes the $k$-th LoRA expert with $r$ ranks, which encapsulates the $k$-th basic ability. These LoRA modules correspond to the singular vectors associated with the top $rK$ largest singular values obtained via SVD, i.e., they capture the most salient components of the original parameter matrix. Meanwhile, $\widehat{\mathbf{W}}$ represents a residual term that accounts for the less important parameters in the pre-trained weights (those associated with relatively small singular values), and we freeze its gradients during the subsequent fine-tuning process. In fact, following the aforementioned SVD-based decoupling, the parameters of different LoRA experts are inherently orthogonal. We provide a formal proof of this property in Appendix A. Consequently, these orthogonal LoRA experts naturally represent mutually orthogonal basic abilities. Based on this decomposition, we obtain a group of initialized LoRA experts. To further align this model with effective MoE-based architectures (Dou et al., 2024; Li et al., 2024a), we introduce learnable routing weights for each expert as follows:

$$\mathbf{W}^0 \xrightarrow{\text{SVD}} \sum_{k=1}^{K} \alpha_k \mathbf{A}_k \mathbf{B}_k + \widehat{\mathbf{W}}, \qquad (7)$$

where the routing weight $\alpha_k$, $k \in \{1, \ldots, K\}$ are initialized to 1 to keep it equal to Eq. (6) and are optimized across different tasks. The routing weights are crucial for enabling the LLM to adaptively select the corresponding basic abilities required for inference on new tasks.

### 3.4. Dynamically Orthogonal Grouping

Since LoRA experts inevitably lose their orthogonality advantage after multi-task instruct-tuning, their nature to represent distinct basic abilities is compromised. To address it, the DOG step is introduced to dynamically regroup the rank-1 components w.r.t. the $rK$ singular values from SVD of each LoRA during training, ensuring that their gradients remain mutually orthogonal throughout the optimization.

Formally, given the dataset $(\mathbf{x}^t, y^t)$ from the $t$-th task, the training gradient of the globally-indexed $i$-th rank-1 LoRA component $[\mathbf{a}_i; \mathbf{b}_i^\top]$, $i \in \{1, \ldots, rK\}$ is formulated as

$$\mathbf{g}_i = \left[ \frac{\partial \mathcal{L}(\mathcal{F}(\mathbf{x}^t; \boldsymbol{\theta}), y^t)}{\partial \mathbf{a}_i}; \frac{\partial \mathcal{L}(\mathcal{F}(\mathbf{x}^t; \boldsymbol{\theta}), y^t)}{\partial \mathbf{b}_i^\top} \right], \quad (8)$$

where $\mathbf{a}_i$ and $\mathbf{b}_i$ denote a specific column of $\mathbf{A}$ and a specific row of $\mathbf{B}$, respectively, corresponding to a particular expert among $K$ experts and associated with one of the $r$ ranks. Given the gradients of these rank-1 components, our goal is to derive a dynamic grouping strategy $\mathbf{\Pi} \in (0,1)^{rK \times K}$ that reassigns the $rK$ rank-1 components into $K$ new rank-$r$ LoRA experts, where each $\pi_{i,k} = 1/0$ denotes whether the original $i$-th LoRA component is assigned / not assigned to the new $k$-th expert. We aim to optimize the strategy $\mathbf{\Pi}$ so that the gradient directions of the resulting $K$ new experts

are as mutually orthogonal as possible, while the gradient directions of the rank-1 components within each expert are similar. This design ensures that each newly formed expert still retains a distinct and orthogonal basic ability.

More specifically, since we focus solely on the directions of gradients as the grouping criterion, we first normalize all gradients onto the unit sphere as follows:

$$\widehat{\mathbf{g}}_i = \mathbf{g}_i \, / \, \|\mathbf{g}_i\|_2. \qquad (9)$$

Then, to minimize the angular deviation among the rank-1 components' gradients within each LoRA expert, we adopt the following objective:

$$\max_{\mathbf{\Pi}} \sum_{k=1}^{K} \sum_{i,j=1}^{rK} \pi_{i,k}\pi_{j,k} \langle \widehat{\mathbf{g}}_i, \widehat{\mathbf{g}}_j \rangle = \max_{\mathbf{\Pi}} \sum_{k=1}^{K} \left\| \sum_{i=1}^{rK} \pi_{i,k}\widehat{\mathbf{g}}_i \right\|^2. \qquad (10)$$

Actually, optimizing this objective also inherently encourages larger, potentially even orthogonal, angular deviation between the gradients of different LoRA experts. A detailed derivation of this objective function and its natural advantages are provided in Appendix B. To implement this objective and further enforce a stronger orthogonality constraint, we design an iterative optimization algorithm. First, we obtain an initial grouping strategy $\mathbf{\Pi}^{(0)}$ by performing spherical $K$-means clustering by optimizing

$$\max_{\mathbf{\Pi}^{(0)}, \mathcal{C}^{(0)}} \sum_{k=1}^{K} \sum_{i=1}^{rK} \pi_{i,k} \left\langle \widehat{\mathbf{g}}_i, \mathbf{c}_k^{(0)} \right\rangle, \ \left\| \mathbf{c}_k^{(0)} \right\| = 1, \qquad (11)$$

where $\mathcal{C}^{(0)} = \{\mathbf{c}_k^{(0)}\}_{k=1}^{K}$ denotes the unit center direction of each cluster. Then, the $\tau$-th iteration of our update algorithm to obtain the optimal $\mathbf{\Pi}$ consists of the following three steps:

**Step 1**: Compute the current centroid vector for each cluster.

$$\mathbf{c}_k^{(\tau)} = \sum_{i=1}^{rK} \pi_{i,k}^{(\tau)} \widehat{\mathbf{g}}_i, \ k \in \{1, \ldots, K\}. \qquad (12)$$

**Step 2**: To orthogonalize these cluster centroids, we perform SVD on the matrix formed by these centroids as

$$\mathbf{C}^{(\tau)} = \left[ \mathbf{c}_1^{(\tau)}, \ldots, \mathbf{c}_K^{(\tau)} \right] = \mathbf{U}_c \mathbf{\Sigma}_c \mathbf{V}_c^\top, \qquad (13)$$

and denote the resulting orthogonal basis as $\mathbf{Q}^{(\tau)} = \mathbf{U}_c \mathbf{V}_c^\top$. Accordingly, the orthogonalized cluster directions, given by the columns $\{\mathbf{q}_k^{(\tau)}\}_{k=1}^{K} = \mathbf{Q}^{(\tau)}$, satisfy the required orthogonality constraint $(\mathbf{q}_k^{(\tau)})^\top \mathbf{q}_k^{(\tau)} = 1$ and $(\mathbf{q}_k^{(\tau)})^\top \mathbf{q}_{l \neq k}^{(\tau)} = 0$.

**Step 3**: Compute the similarity $\langle \widehat{\mathbf{g}}_i, \mathbf{q}_k^{(\tau)} \rangle$ between the gradient and the orthogonalized cluster directions, and solve the following integer optimization problem to obtain the updated policy $\mathbf{\Pi}^{(\tau)}$:

$$\max_{\mathbf{\Pi}^{(\tau)}} \sum_{k=1}^{K} \sum_{i=1}^{rK} \pi_{i,k}^{(\tau)} \langle \widehat{\mathbf{g}}_i, \mathbf{q}_k^{(\tau)} \rangle, \ \sum_{i=1}^{rK} \pi_{i,k}^{(\tau)} = r. \qquad (14)$$

The iterative algorithm terminates when the strategy $\Pi$ stabilizes (i.e., no longer changes) or after a maximum of 10 optimization steps. Using the final strategy $\Pi$, we regroup the rank-1 LoRA components accordingly. Notably, the output of the MoE architecture remains invariant after our regrouping, even though each expert possesses its own routing weights $\alpha$, a fact that is proven in Appendix C.

## 4. Experiments

In this section, we implement experiments to demonstrate the performance of our proposed BADIT. More experimental results, including intra- and inter-expert gradient angles, sensitivity analysis, and detailed performance across different tasks, are provided in Appendix H.

### 4.1. Experimental Settings

**Datasets and evaluation.** We evaluate our BADIT method on *SuperNI* (Wang et al., 2022b), a multi-task instruct-tuning dataset comprising 15 distinct natural language tasks, e.g., sentiment analysis, each accompanied by its own training and evaluation sets. Detailed statistics and descriptions of the dataset are provided in Appendix G.1. During training, we assess our approach under two multi-task training paradigms: *mixed training* and *sequential training*. In mixed training, we mix the training sets of all 15 tasks and train them simultaneously. In sequential training, we train the tasks sequentially according to a prescribed order, and we report the average performance across five different task orders to mitigate the influence of ordering.

**Foundation LLMs.** We validate our method on 6 instruct-tuned LLMs spanning different LLM families and sizes, including Qwen3-8B, Qwen3-4B (Yang et al., 2025a), Llama3-8B, Llama3-3B (AI@Meta, 2024), Gemma2-9B, and Gemma2-2B (Rivière et al., 2024). Their detailed model cards are provided in Appendix G.2.

**Baselines.** We compare our approach against two LoRA series methods: LoRA (Hu et al., 2022) and PiSSA (Meng et al., 2024), and three MoE-based methods specifically designed for multi-task instruct-tuning: OLoRA (Wang et al., 2023a), LoRAMoE (Dou et al., 2024), and Gain-LoRA (Liang et al., 2025). Detailed descriptions of these methods are provided in Appendix G.3.

In addition to the experimental setup described above, implementation details of our method and the precise formulations of the evaluation metrics we use can be found in Appendices G.4 and G.5, respectively.

### 4.2. Main Results

Our main experimental results are presented in Table 1, where each experiment is repeated 5 times with different random seeds, and we report the average performance and its corresponding standard deviations. Notably, under the sequential training setting, each random seed also determines a different task order; consequently, the reported standard deviations in this setting are generally larger due to the added sensitivity to task sequencing. Among the evaluation metrics we employ, ROUGE denotes the average performance across all tasks. Forward measure the performance gain achieved by multi-task training compared to training each task individually. Forget Rate and Backward quantify the impact on previously learned tasks when training on new tasks for the sequential training setting. The formal definitions of these metrics are provided in Appendix G.5.

Generally, our method consistently outperforms existing multi-task instruct-tuning approaches across all foundation LLMs and evaluation metrics. For example, BADIT achieves an average improvement of approximately 2.68 points over the state-of-the-art method, GainLoRA, on the ROUGE metric. This not only demonstrates that our approach effectively enhances the multi-task performance of LLMs but also shows its ability to substantially mitigate interference among tasks and alleviate catastrophic forgetting in the sequential training scenario.

### 4.3. Ablation Study

We conduct an ablation study in Table 2 to investigate the impact of removing two key components: BAD and DOG. Specifically, **w/o BAD** means we start from a LoRAMoE initialized with zero vectors and enforce orthogonality among rank-1 LoRA components only during training, without the SVD initialization. **w/o DOG** means we apply SVD-based decoupled initialization but do not enforce orthogonality among experts during training.

The experiments are carried out on Qwen3-8B, Llama3-8B, and Gemma-9B, and we report ROUGE scores under both mixed and sequential training settings. Generally, removing either component consistently degrades LLM performance, underscoring the critical role both modules play in enhancing multi-task capability and mitigating interference among tasks. Moreover, we observe that the DOG module generally contributes more substantially: while SVD-based initialization alone ensures that distinct LoRA components remain orthogonal, thus represent diverse basic abilities, only during the early stages of training, this orthogonality gradually erodes as training progresses. Consequently, dynamically preserving orthogonality throughout training (as enabled by DOG) proves essential for sustaining performance gains and robustness in two multi-task settings.

### 4.4. Intra- and Inter-Expert Gradient Angles

To investigate whether our method can dynamically promote orthogonality among expert gradients through clustering,

*Table 1.* Experimental results of BADIT under two implementation setting: mixed training and sequential training, respectively. We repeat experiments with 5 different seeds, and report their average scores and standard deviations.

| | Method | Mixed Training | | Sequential Training | | | |
|---|---|---|---|---|---|---|---|
| | | ROUGE↑ | Forward↑ | ROUGE↑ | Forget Rate↓ | Forward↑ | Backward↑ |
| **Qwen3-8B** | LoRA (Hu et al., 2022) | 54.22±0.72 | -1.66±0.73 | 47.08±1.95 | 9.21±2.24 | -0.11±0.83 | -8.11±2.19 |
| | OLoRA (Wang et al., 2023a) | 54.53±0.46 | -0.45±0.46 | 48.57±3.15 | 8.73±3.18 | -1.15±0.39 | -6.78±2.95 |
| | LoRAMoE (Dou et al., 2024) | 54.64±0.71 | -0.57±0.71 | 48.07±0.79 | 8.47±0.59 | -1.26±0.44 | -6.23±0.55 |
| | PiSSA (Meng et al., 2024) | 53.57±0.35 | -2.30±0.35 | 47.69±2.52 | 7.50±2.80 | -0.06±0.79 | -6.83±2.32 |
| | GainLoRA (Liang et al., 2025) | 54.33±0.73 | -1.78±0.73 | 48.44±1.02 | 8.96±0.95 | -0.73±1.53 | -6.42±0.71 |
| | ★ BADIT (ours) | **55.87**±0.35 | **0.92**±0.35 | **50.86**±1.42 | **6.95**±1.57 | **0.44**±0.53 | **-5.43**±1.47 |
| **Qwen3-4B** | LoRA (Hu et al., 2022) | 53.31±0.92 | -1.24±0.92 | 48.54±1.63 | 8.53±1.58 | -1.31±0.37 | -6.68±1.34 |
| | OLoRA (Wang et al., 2023a) | 53.50±0.76 | 0.15±0.76 | 47.83±1.54 | 8.78±2.90 | -1.77±0.61 | -6.47±2.66 |
| | LoRAMoE (Dou et al., 2024) | 53.63±0.25 | -0.68±0.25 | 48.58±1.79 | 8.50±1.59 | -0.74±0.44 | -6.25±1.55 |
| | PiSSA (Meng et al., 2024) | 52.87±0.70 | -0.45±0.70 | 48.26±1.79 | 8.05±1.44 | -0.64±0.67 | -6.12±1.64 |
| | GainLoRA (Liang et al., 2025) | 53.24±0.72 | -1.33±0.72 | 48.07±1.70 | 8.47±1.50 | -1.26±0.44 | -6.23±1.75 |
| | ★ BADIT (ours) | **55.22**±0.42 | **0.91**±0.42 | **50.74**±2.27 | **5.95**±2.12 | **0.62**±0.70 | **-4.21**±2.27 |
| **Llama3-8B** | LoRA (Hu et al., 2022) | 52.58±0.79 | -1.02±0.79 | 44.88±3.61 | 12.41±3.49 | -1.90±1.48 | -10.23±3.63 |
| | OLoRA (Wang et al., 2023a) | 52.89±0.89 | -0.41±0.89 | 45.25±2.84 | 11.45±3.04 | 0.00±0.77 | -10.01±2.88 |
| | LoRAMoE (Dou et al., 2024) | 52.95±0.47 | -1.96±0.47 | 45.47±3.68 | 11.78±3.25 | 0.27±0.93 | -9.71±3.04 |
| | PiSSA (Meng et al., 2024) | 52.88±0.97 | -0.72±0.97 | 45.23±2.81 | 12.54±3.07 | -0.63±0.61 | -10.67±3.04 |
| | GainLoRA (Liang et al., 2025) | 52.71±1.07 | -2.13±1.07 | 45.04±2.78 | 12.70±2.63 | -1.17±0.85 | -10.94±3.07 |
| | ★ BADIT (ours) | **54.75**±0.69 | **0.79**±0.69 | **48.83**±1.89 | **8.57**±1.75 | **1.71**±1.53 | **-3.49**±2.67 |
| **Llama3-3B** | LoRA (Hu et al., 2022) | 50.07±0.97 | -0.23±0.96 | 45.14±3.05 | 9.79±2.17 | -0.17±0.90 | -4.91±2.91 |
| | OLoRA (Wang et al., 2023a) | 49.83±0.70 | -0.47±0.69 | 45.59±3.10 | 10.07±3.01 | -0.74±0.91 | -5.95±2.52 |
| | LoRAMoE (Dou et al., 2024) | 49.72±0.52 | -0.59±0.51 | 46.20±2.90 | 8.58±2.82 | -0.59±1.38 | -3.51±2.63 |
| | PiSSA (Meng et al., 2024) | 49.64±0.60 | -0.27±0.59 | 45.49±2.95 | 9.13±2.34 | -0.93±1.00 | -5.27±2.39 |
| | GainLoRA (Liang et al., 2025) | 49.60±0.60 | -0.23±0.60 | 45.34±3.14 | 9.67±2.63 | -0.78±1.23 | -5.66±3.24 |
| | ★ BADIT (ours) | **51.45**±0.72 | **0.61**±0.72 | **48.13**±2.26 | **6.25**±2.45 | **0.71**±0.85 | **-2.57**±2.86 |
| **Gemma2-9B** | LoRA (Hu et al., 2022) | 55.16±0.49 | -1.35±0.49 | 46.80±2.01 | 10.46±2.05 | -0.32±0.79 | -9.38±2.10 |
| | OLoRA (Wang et al., 2023a) | 54.78±0.81 | -1.05±0.81 | 46.05±1.50 | 11.20±1.64 | -0.69±0.69 | -13.57±1.28 |
| | LoRAMoE (Dou et al., 2024) | 54.78±0.64 | -0.91±0.62 | 46.98±2.89 | 11.99±2.99 | 0.17±0.45 | -10.11±2.76 |
| | PiSSA (Meng et al., 2024) | 54.20±1.07 | -1.78±1.07 | 44.93±3.09 | 12.27±3.26 | -2.24±0.27 | -14.80±3.10 |
| | GainLoRA (Liang et al., 2025) | 54.29±0.45 | -1.88±0.45 | 46.51±2.22 | 12.62±2.77 | -0.62±0.48 | -10.80±2.67 |
| | ★ BADIT (ours) | **57.35**±0.67 | **0.54**±0.67 | **48.34**±2.11 | **10.31**±2.22 | **0.39**±0.80 | **-8.45**±1.46 |
| **Gemma2-2B** | LoRA (Hu et al., 2022) | 51.72±0.96 | -1.30±0.96 | 43.16±3.06 | 13.21±3.42 | -0.03±1.77 | -11.28±2.94 |
| | OLoRA (Wang et al., 2023a) | 51.82±0.71 | -0.87±0.71 | 44.93±3.12 | 10.94±3.36 | -1.03±0.31 | -10.55±2.61 |
| | LoRAMoE (Dou et al., 2024) | 51.36±0.36 | -1.05±0.37 | 43.88±2.68 | 11.66±3.12 | -0.86±0.17 | -9.95±2.69 |
| | PiSSA (Meng et al., 2024) | 50.96±0.44 | -1.28±0.44 | 42.57±2.90 | 12.74±3.59 | -1.65±0.62 | -10.57±3.37 |
| | GainLoRA (Liang et al., 2025) | 51.26±0.76 | -1.70±0.76 | 44.16±2.36 | 11.81±2.38 | -0.92±0.57 | -10.27±2.63 |
| | ★ BADIT (ours) | **53.60**±0.36 | **0.70**±0.36 | **46.76**±2.16 | **9.30**±2.98 | **0.25**±0.49 | **-7.65**±2.86 |

we visualize in Fig. 4 the evolution of two angular metrics across training epochs: (1) the inter-expert gradient angle, i.e., the angle between gradients of different experts, and (2) the intra-expert gradient angle, i.e., the angle between gradients of different rank-1 components within the same expert. The results reveal that, in contrast to LoRAMoE, which lacks dynamic regrouping, our method consistently maintains inter-expert gradient angles near 90° throughout training, indicating sustained orthogonality between experts. Meanwhile, the intra-expert gradient angles in our approach remain around 60°, approximately 20° lower than those observed in LoRAMoE. Therefore, these findings demonstrate that our method effectively preserves orthogonality among expert gradients across the entire training process, thereby enabling a more precise and disentangled representation of basic abilities. Additional results across more LLMs are

provided in Appendix H.2.

### 4.5. Computation Budgets

In this section, we analyze the computational overhead of our proposed two-stage method. A formal $\mathcal{O}$ time complexity analysis is provided in Appendix E. Specifically, Table 3 reports, across 6 LLMs, the relative increase in computational time cost of our method compared to LoRAMoE, under identical settings of expert count and LoRA rank. Overall, our method incurs approximately 1.22× the training time of LoRAMoE, with the majority of this additional cost attributed to the DOG stage. This observation leads to two key insights: First, despite involving an SVD-based decoupling step, our BAD stage does not increase, and may even reduce, computational time due to faster convergence

*Table 2.* Ablation study of the BAD and DOG methods in BADIT. ST R., ST F., and MT R. indicate the ROUGE and forget rate under the sequential training, and the ROUGE under the mixed training, respectively. Three LLMs use their 8B and 9B versions.

| | Method | ST R. | ST F. | MT R. | Δ |
|---|---|---|---|---|---|
| **Qwen3** | BADIT | 50.86 | 6.95 | 55.87 | - |
| | w/o BAD | 50.07 | 7.34 | 55.02 | **2.03** |
| | w/o DOG | 49.70 | 8.20 | 54.85 | **3.43** |
| | w/o BAD & DOG | 48.07 | 8.47 | 54.64 | **5.54** |
| **Llama3** | BADIT | 48.83 | 8.57 | 54.75 | - |
| | w/o BAD | 47.92 | 9.30 | 54.02 | **2.37** |
| | w/o DOG | 47.33 | 9.74 | 53.74 | **3.68** |
| | w/o BAD & DOG | 45.47 | 11.78 | 52.95 | **8.37** |
| **Gemma2** | BADIT | 48.34 | 10.31 | 57.35 | - |
| | w/o BAD | 47.78 | 10.57 | 56.71 | **1.46** |
| | w/o DOG | 47.60 | 11.07 | 56.03 | **2.82** |
| | w/o BAD & DOG | 46.98 | 11.99 | 54.78 | **5.61** |

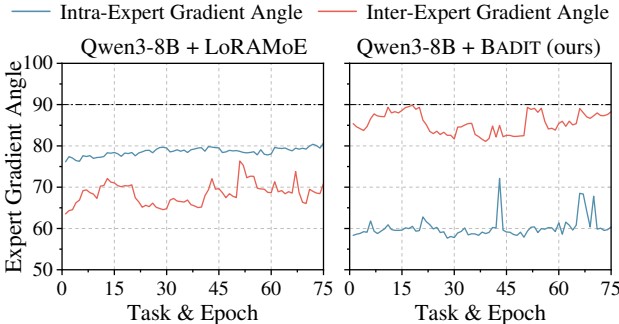

*Figure 4.* Intra- and inter-expert gradient angles across epochs.

during training. Second, the DOG stage introduces extra overhead because it performs spherical $K$-means clustering and integer optimization on parameter gradients, which are computationally intensive operations primarily executed on CPUs. Although our approach entails a modest increase in training time, the resulting substantial gains in model performance render this trade-off well justified and acceptable.

## 5. Related Work

Cross-task interference is a critical challenge in the multi-task instruct-tuning of LLMs (Luo et al., 2023; Jang et al., 2023). Generally, existing approaches can be broadly classified into three categories (Wang et al., 2024a; Zhou et al., 2024): *rehearsal-based* methods (Huang et al., 2024a; Wang et al., 2024b) store and replay historical data samples; *optimization-based* methods (Li et al., 2024b; Jiang et al., 2025) introduce training regularizations, e.g., orthogonal constraints (Wang et al., 2023a), to guide the optimization direction; and *architecture-based* methods (Dou et al., 2024; Zhao et al., 2024a; Yang et al., 2025b) employ parameter-efficient modules designed to isolate task-specific knowledge and mitigate interference across tasks.

In this section, we focus primarily on architecture-based

*Table 3.* Time cost of different methods across 6 LLMs under the sequential training setting.

| Method | LoRA | LoRAMoE | PiSSA | BADIT | - DOG | - BAD |
|---|---|---|---|---|---|---|
| Qwen3-8B | 0.92× | 1× | 1.18× | 1.25× | 1.14× | 1.24× |
| Qwen3-4B | 0.75× | 1× | 0.86× | 1.18× | 0.96× | 1.20× |
| Llama3-8B | 0.68× | 1× | 0.84× | 1.16× | 0.95× | 1.15× |
| Llama3-3B | 0.71× | 1× | 0.61× | 1.27× | 0.96× | 1.27× |
| Gemma2-9B | 0.74× | 1× | 1.08× | 1.25× | 1.02× | 1.24× |
| Gemma2-2B | 0.79× | 1× | 0.69× | 1.24× | 1.01× | 1.24× |
| Average | **0.76×** | **1×** | **0.87×** | **1.22×** | **1.00×** | **1.22×** |

methods, and more comprehensive literature reviews can be seen in Appendix F. Specifically, the basic idea of architecture-based methods is to mitigate cross-task interference by explicitly assigning task-specific, independent trainable parameters. For example, Wang et al. (2023b) allocate a small set of private parameters, e.g., soft prompts (Li & Liang, 2021), and train them alongside a shared pre-trained model. Similarly, SAPT (Zhao et al., 2024b) addresses knowledge transfer among private parameters by introducing a shared attention module. More recently, several works have alleviated interference by combining LoRA (Hu et al., 2022) with MoE architectures (Shazeer et al., 2017). These methods design a LoRA-based expert for each task within an MoE framework and employ a routing mechanism to automatically select one or more appropriate experts for incoming tasks (Dou et al., 2024; Ma et al., 2024; Feng et al., 2024; Li et al., 2024a; Liu et al., 2024b; Chen et al., 2024). Our approach builds upon this paradigm by dynamically adapting the expert models so that each expert encapsulates a distinct and orthogonal basic ability.

## 6. Conclusion

In this work, we identify a fundamental limitation in existing multi-task instruct-tuning methods: despite efforts to isolate task-specific parameters, substantial parameter sharing persists across tasks, leading to unresolved cross-task interference. Through empirical analysis, we reveal that LLMs consistently co-activate groups of parameters, suggesting that LLMs encode a small set of orthogonal basic abilities, each reusable across multiple tasks. Motivated by this insight, we propose BADIT, which decomposes LLM parameters into high-singular-value LoRA experts representing these basic abilities. By dynamically enforcing orthogonality among these experts via spherical clustering of their rank-1 components during training, BADIT effectively mitigates gradient conflict while enabling flexible task representation as linear combinations of shared abilities. Extensive experiments on the *SuperNI* benchmark across six LLMs demonstrate that BADIT outperforms state-of-the-art methods by an average of 2.68 ROUGE points and exhibits significantly reduced cross-task interference.

## Acknowledgements

This work was supported in part by the National Natural Science Foundation of China (No.62276113) and JST ASPIRE Grant Number JPMJAP2405.

## Impact Statement

We propose a novel approach for multi-task instruct-tuning of LLMs, motivated by the observation that prior methods still rely on shared parameters, which can lead to conflicting gradients across tasks. To address this, we decompose the basic abilities inherent in the LLM itself to represent any task. We validate our method using publicly available datasets spanning 15 NLP tasks, e.g., sentiment analysis, none of which contain harmful, private, or sensitive content. Furthermore, our experiments are conducted on open-source instruct-tuned LLMs, thereby avoiding any copyright risks.

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

## A. Proof of the Orthogonality of SVD Initialization

In this section, we formally prove that the proposed SVD-based initialization of BAD yields mutually orthogonal low-rank subspaces across experts at initialization.

**Theorem A.1.** *Let $\mathbf{W} \in \mathbb{R}^{m \times n}$ be a pre-trained LLM weight matrix, and let*

$$\mathbf{W} = \mathbf{U}\mathbf{\Sigma}\mathbf{V}^\top = \sum_{i=1}^{\min(m,n)} \sigma_i \mathbf{u}_i \mathbf{v}_i^\top$$

*be its compact singular value decomposition, where $\sigma_1 \geq \sigma_2 \geq \ldots \geq \sigma_{\min(m,n)}$ are singular values, and $\mathbf{u}_i \in \mathbb{R}^m$, $\mathbf{v}_i \in \mathbb{R}^n$ are the corresponding orthonormal left and right singular vectors.*

*Then the rank-1 matrices $\mathbf{u}_i \mathbf{v}_i^\top$ form an orthogonal basis under the Frobenius inner product as*

$$\left\langle \mathbf{u}_i \mathbf{v}_i^\top, \mathbf{u}_j \mathbf{v}_j^\top \right\rangle_F = \mathrm{Tr}\left( \left(\mathbf{u}_i \mathbf{v}_i^\top\right)^\top \mathbf{u}_j \mathbf{v}_j^\top \right) \tag{15}$$
$$= \left(\mathbf{u}_i^\top \mathbf{u}_j\right)\left(\mathbf{v}_i^\top \mathbf{v}_j\right) = \delta_{ij},$$

*where $\delta_{ij} = 1$ if $i = j$ and $0$ otherwise.*

*Proof.* The orthonormality of $\mathbf{u}_i$ and $\mathbf{v}_i$ from the SVD ensures $\mathbf{u}_i^\top \mathbf{u}_j = \delta_{ij}$ and $\mathbf{v}_i^\top \mathbf{v}_j = \delta_{ij}$. Substituting into the definition of Frobenius inner product directly yields Eq. (15), proving that $\mathbf{u}_i \mathbf{v}_i^\top$ are orthogonal in the Frobenius sense. $\square$

**Proposition A.2** (Orthogonal LoRA Expert Initialization)**.**
*Let $R = rK \ll \min(m, n)$ be the total SVD rank used for initializing $K$ experts, each of rank $r$. Partition the first $R$ SVD components into $K$ contiguous, non-overlapping blocks: the $k$-th expert ($k \in \{0, \ldots, K-1\}$) receives indices from $rk$ to $r(k+1) - 1$. Define*

$$\mathbf{A}_k = \mathbf{U}_k \sqrt{\frac{\mathbf{\Sigma}_k}{\alpha}}, \quad \mathbf{B}_k = \sqrt{\frac{\mathbf{\Sigma}_k}{\alpha}} \mathbf{V}_k^\top,$$

*where $\mathbf{U}_k \in \mathbb{R}^{m \times r}$ and $\mathbf{V}_k \in \mathbb{R}^{n \times r}$ contain the assigned singular vectors and $\mathbf{\Sigma}_k = \mathrm{diag}(\sigma_{rk}, \ldots, \sigma_{r(k+1)-1})$. The LoRA scaling factor is $\alpha$ (scaling $s = \alpha/r$). The initialization update for expert $k$ is*

$$\Delta\mathbf{W}_k = s\mathbf{A}_k\mathbf{B}_k = \mathbf{U}_k\mathbf{\Sigma}_k\mathbf{V}_k^\top = \sum_{i=rk}^{r(k+1)-1} \sigma_i \mathbf{u}_i \mathbf{v}_i^\top.$$

*Proof.* For two distinct experts $k \neq l$, their updates have disjoint index sets as

$$\Delta\mathbf{W}_k = \sum_{i \in \mathcal{I}_k} \sigma_i \mathbf{u}_i \mathbf{v}_i^\top, \quad \Delta\mathbf{W}_l = \sum_{j \in \mathcal{I}_l} \sigma_j \mathbf{u}_j \mathbf{v}_j^\top,$$

where $\mathcal{I}_k \cap \mathcal{I}_l = \varnothing$. By Theorem A.1, for $i \in \mathcal{I}_k$, $j \in \mathcal{I}_l$, we have $\langle \mathbf{u}_i \mathbf{v}_i^\top, \mathbf{u}_j \mathbf{v}_j^\top \rangle_F = 0$. Therefore,

$$\langle \Delta\mathbf{W}_k, \Delta\mathbf{W}_l \rangle_F = \sum_{i \in I_k} \sum_{j \in I_l} \sigma_i \sigma_j \langle \mathbf{u}_i \mathbf{v}_i^\top, \mathbf{u}_j \mathbf{v}_j^\top \rangle_F = 0.$$

Thus, all initialized experts are mutually orthogonal under the Frobenius inner product. $\square$

*Remark A.3.* This orthogonality ensures non-redundant and diverse low-rank subspaces across experts at initialization, and the construction preserves the SVD's optimal information content for the chosen rank $R$.

## B. Detailed Regrouping Objective

Given gradients $\{\widehat{\mathbf{g}}_i\}_{i=1}^{rK}$, the grouping aims to optimize the strategy $\mathbf{\Pi} \in (0, 1)^{rK \times K}$, where each $\pi_{i,k} = 1/0$ denotes whether the original $i$-th component is assigned / not assigned to the new $k$-th expert. For these fixed gradients, we aim to maximize the sum of inner products within groups while simultaneously minimizing the sum of inner products between groups. To this end, we formulate the following objective function:

$$\max_{\mathbf{\Pi}} \mathcal{L}_{\mathrm{intra}} - \lambda \mathcal{L}_{\mathrm{inter}}, \tag{16}$$

$$\mathcal{L}_{\mathrm{intra}} = \sum_{k=1}^{K} \sum_{i,j=1}^{rK} \pi_{i,k} \pi_{j,k} \langle \widehat{\mathbf{g}}_i, \widehat{\mathbf{g}}_j \rangle,$$

$$\mathcal{L}_{\mathrm{inter}} = \sum_{l \neq k} \sum_{i,j=1}^{rK} \pi_{i,k} \pi_{j,l} \langle \widehat{\mathbf{g}}_i, \widehat{\mathbf{g}}_j \rangle,$$

where $\lambda > 0$ is a trade-off hyper-parameter. The first term of the objective function encourages smaller angular differences (i.e., higher cosine distance) among gradients within each expert, while the second term promotes larger angular separation (i.e., lower cosine distance) between gradients of different experts. Meanwhile, due to

$$\sum_{i,j=1}^{rK} \pi_{i,k} \pi_{j,k} \langle \widehat{\mathbf{g}}_i, \widehat{\mathbf{g}}_j \rangle = \left\| \sum_{i=1}^{rK} \pi_{i,k} \widehat{\mathbf{g}}_i \right\|^2 .$$

Therefore, the objective is equivalent to

$$\mathcal{L}_{\text{intra}} = \sum_{k=1}^{K} \left\| \sum_{i=1}^{rK} \pi_{i,k} \widehat{\mathbf{g}}_i \right\|^2 ,$$

$$\mathcal{L}_{\text{inter}} = \left\| \sum_{i=1}^{rK} \widehat{\mathbf{g}}_i \right\|^2 - \sum_{k=1}^{K} \left\| \sum_{i=1}^{rK} \pi_{i,k} \widehat{\mathbf{g}}_i \right\|^2 .$$

Moreover, regardless of how the gradients are grouped, the total inner product across all pairs of gradients remains constant as

$$\sum_{i,j=1}^{rK} \pi_{i,k} \pi_{j,k} \langle \widehat{\mathbf{g}}_i, \widehat{\mathbf{g}}_j \rangle = \left\| \sum_{i=1}^{rK} \widehat{\mathbf{g}}_i \right\|^2 = \mathcal{L}_{\text{intra}} + \mathcal{L}_{\text{inter}} = C.$$

Accordingly, the objective in Eq. (16) can be formulated as:

$$\max_{\mathbf{\Pi}} \mathcal{L}_{\text{intra}} - \lambda (C - \mathcal{L}_{\text{intra}}) = (1 + \lambda)\mathcal{L}_{\text{intra}} - \lambda C,$$

$$\Rightarrow \max_{\mathbf{\Pi}} (1 + \lambda) \sum_{k=1}^{K} \left\| \sum_{i=1}^{rK} \pi_{i,k} \widehat{\mathbf{g}}_i \right\|^2 - \lambda C,$$

$$\Rightarrow \max_{\mathbf{\Pi}} \sum_{k=1}^{K} \left\| \sum_{i=1}^{rK} \pi_{i,k} \widehat{\mathbf{g}}_i \right\|^2 ,$$

which is equivalent to the objective in Eq. (10).

## C. Proof of Regrouping Invariance

### C.1. Invariance of Rank-1 Regrouping

We consider a LoRAMoE with $K$ experts, each expert $k$ parameterized by a low-rank adaptation of rank $r$. Denote by $\mathbf{A}_k \in \mathbb{R}^{m \times r}$ and $\mathbf{B}_k \in \mathbb{R}^{r \times n}$ the LoRA factors for expert $k$, so that the LoRA update for expert $k$ is given by

$$\Delta \mathbf{W}_k = \mathbf{A}_k \mathbf{B}_k = \sum_{i=1}^{r} \mathbf{a}_{ki} \mathbf{b}_{ki}^\top ,$$

where $\mathbf{a}_{ki} \in \mathbb{R}^m$ is the $i$-th column of $\mathbf{A}_k$ and $\mathbf{b}_{ki} \in \mathbb{R}^n$ is the $i$-th row of $\mathbf{B}_k$.

Let the residual weight matrix of the layer be $\widehat{\mathbf{W}}$. The total weight matrix of the LoRAMoE layer is

$$\mathbf{W} = \widehat{\mathbf{W}} + \sum_{k=1}^{K} \Delta \mathbf{W}_k .$$

Define the set of all $rK$ rank-1 components as

$$\mathcal{R} = \{ \mathbf{R}_i = \mathbf{a}_i \mathbf{b}_i^\top \mid i = 1, 2, \dots, rK \},$$

where each $\mathbf{R}_i \in \mathbb{R}^{m \times n}$ is exactly one rank-1 term from some expert's LoRA factors.

**Theorem C.1** (Invariance of Rank-1 Regrouping). *Suppose the collection $\mathcal{R}$ is fixed element-wise (i.e., each $\mathbf{R}_i$ is unchanged in value), and that the set $\mathcal{R}$ is partitioned into $K$ disjoint groups of size $r$ to form the per-expert LoRA updates. If the partition is changed to any other partition into $K$ disjoint groups of size $r$ without altering the individual $\mathbf{R}_i$ values, then the total LoRAMoE weight matrix $\mathbf{W}$ remains unchanged.*

*Proof.* Let the original partition of indices $\{1, \dots, rK\}$ be $\mathcal{G}_1^{\text{old}}, \dots, \mathcal{G}_K^{\text{old}}$, with $|\mathcal{G}_k^{\text{old}}| = r$ for all $k$ and $\bigcup_{k=1}^{K} \mathcal{G}_k^{\text{old}} = \{1, \dots, rK\}$, $\mathcal{G}_k^{\text{old}} \cap \mathcal{G}_j^{\text{old}} = \varnothing$ for $k \neq j$.

The original total LoRA update is

$$\Delta \mathbf{W}_{\text{old}} = \sum_{k=1}^{K} \sum_{i \in \mathcal{G}_k^{\text{old}}} \mathbf{R}_i = \sum_{i=1}^{rK} \mathbf{R}_i . \tag{17}$$

Consider any new partition $\mathcal{G}_1^{\text{new}}, \dots, \mathcal{G}_K^{\text{new}}$ of $\{1, \dots, rK\}$, with the same cardinality constraint $|\mathcal{G}_k^{\text{new}}| = r$ and forming a disjoint union of the entire index set.

The new total LoRA update is

$$\Delta \mathbf{W}_{\text{new}} = \sum_{k=1}^{K} \sum_{i \in \mathcal{G}_k^{\text{new}}} \mathbf{R}_i = \sum_{i=1}^{rK} \mathbf{R}_i . \tag{18}$$

Comparing Eqs. (17) and (18) yields

$$\Delta \mathbf{W}_{\text{new}} = \Delta \mathbf{W}_{\text{old}} .$$

Therefore, since $\mathbf{W} = \widehat{\mathbf{W}} + \Delta \mathbf{W}$, we have $\mathbf{W}^{\text{new}} = \mathbf{W}^{\text{old}}$. $\square$

*Remark* C.2. This invariance result assumes that the MoE output is computed by adding *all* experts' weight updates to $\mathbf{W}$ simultaneously. If, as in typical MoE inference, a gating network selects only a subset of experts for a given input, the per-expert matrices may change under re-grouping, and the overall functional behavior may differ despite the equality of the summed update $\sum_{i=1}^{rK} \mathbf{R}_i$.

### C.2. Invariance of Gated Rank-1 Regrouping

For a given input $\mathbf{x}$, a gating network produces scalar coefficients $g_k(\mathbf{x}) \in \mathbb{R}$ for $k \in \{1, \dots, K\}$. The output of the MoE layer is

$$\mathbf{h}(\mathbf{x}) = \sum_{k=1}^{K} g_k(\mathbf{x}) \left[ \left( \widehat{\mathbf{W}} + \Delta \mathbf{W}_k \right) \mathbf{x} \right] .$$

We consider the LoRAMoE setting and notation from the previous theorem as

$$\Delta \mathbf{W}_k = \sum_{i \in \mathcal{G}_k} \mathbf{R}_i, \quad \mathbf{R}_i = \mathbf{a}_i \mathbf{b}_i^\top,$$

where expert $k$ has gating coefficient $g_k(\mathbf{x})$ for input $\mathbf{x}$.

Suppose, for a given $\mathbf{x}$, we transfer a single rank-1 component $\mathbf{M} = \mathbf{R}_{i^*}$ from its original expert $\mathbf{u} = \mathbf{u}(i^*)$ to a new expert $\mathbf{v} = \mathbf{v}(i^*)$, with $\mathbf{u} \neq \mathbf{v}$. We rescale it by

$$\gamma(\mathbf{x}) = \frac{g_u(\mathbf{x})}{g_v(\mathbf{x})}, \quad \text{assuming } g_v(\mathbf{x}) \neq 0.$$

Define the modified expert LoRA updates (for this $\mathbf{x}$) as

$$\Delta \widehat{\mathbf{W}}_u = \Delta \mathbf{W}_u - \mathbf{M},$$
$$\Delta \widehat{\mathbf{W}}_v = \Delta \mathbf{W}_v + \gamma(\mathbf{x}) \mathbf{M},$$
$$\Delta \widehat{\mathbf{W}}_k = \Delta \mathbf{W}_k \text{ for } k \notin \{\mathbf{u}, \mathbf{v}\}.$$

**Theorem C.3** (Invariance of Gated Rank-1 Regrouping). *Under the above modification, the total gated low-rank adaptation for this $x$ satisfies*

$$\sum_{k=1}^{K} g_k(\mathbf{x}) \, \Delta \widehat{\mathbf{W}}_k = \sum_{k=1}^{K} g_k(\mathbf{x}) \, \Delta \mathbf{W}_k.$$

*Proof.* Consider the modified total (for fixed $\mathbf{x}$)

$$\sum_{k=1}^{K} g_k(\mathbf{x}) \Delta \widehat{\mathbf{W}}_k = g_u(\mathbf{x}) \Delta \widehat{\mathbf{W}}_u + g_v(\mathbf{x}) \, \Delta \widehat{\mathbf{W}}_v$$
$$+ \sum_{k \notin \{\mathbf{u}, \mathbf{v}\}} g_k(\mathbf{x}) \, \Delta \mathbf{W}_k$$
$$= g_u(\mathbf{x}) \left( \Delta \mathbf{W}_u - \mathbf{M} \right) + g_v(\mathbf{x}) \left( \Delta \mathbf{W}_v + \gamma(\mathbf{x}) \, \mathbf{M} \right)$$
$$+ \sum_{k \notin \{\mathbf{u}, \mathbf{v}\}} g_k(\mathbf{x}) \, \Delta \mathbf{W}_k$$
$$= \left[ \sum_{k=1}^{K} g_k(\mathbf{x}) \, \Delta \mathbf{W}_k \right] + \left[ -g_u(\mathbf{x}) + g_v(\mathbf{x}) \, \gamma(\mathbf{x}) \right] \mathbf{M}.$$

Now substitute $\gamma(\mathbf{x}) = g_u(\mathbf{x})/g_v(\mathbf{x})$

$$-g_u(\mathbf{x}) + g_v(\mathbf{x}) \frac{g_u(\mathbf{x})}{g_v(\mathbf{x})} = -g_u(\mathbf{x}) + g_u(\mathbf{x}) = 0.$$

Thus the extra term vanishes, and

$$\sum_{k=1}^{K} g_k(\mathbf{x}) \, \Delta \widehat{\mathbf{W}}_k = \sum_{k=1}^{K} g_k(\mathbf{x}) \, \Delta \mathbf{W}_k.$$

This proves invariance for this fixed $\mathbf{x}$. $\square$

---

**Algorithm 1** The overall training pipeline of BADIT

**Input:** Pre-trained LLM weights $\boldsymbol{\theta}^0$, number of experts $K$, rank per expert $r$
**Output:** Fine-tuned model with orthogonal basic abilities
  ▷ *Basic Ability Decomposition*
  **for** each weight matrix $\mathbf{W}^0 \in \boldsymbol{\theta}^0$ **do**
    Decompose $\mathbf{W}^0 \to \sum_{k=1}^{K} \mathbf{A}_k \mathbf{B}_k + \widehat{\mathbf{W}}$ using Eq. (6)
    Freeze residual $\widehat{\mathbf{W}}$ during fine-tuning
  **end for**
  ▷ *Dynamically Orthogonal Grouping*
  **for** each training batch $(\mathbf{x}^t, y^t)$ **do**
    Compute gradients $\widehat{\mathbf{g}}_i$ using Eqs. (8) and (9)
    Initialize grouping via spherical $K$-means using Eq. (11)
    **for** $\tau = 1$ to $10$ **do**
      Compute cluster centroids using Eq. (12)
      Orthogonalize centroids via SVD using Eq. (13)
      Update assignment $\mathbf{\Pi}^{(\tau)}$ using Eq. (14)
      **if** $\mathbf{\Pi}^{(\tau)}$ is unchanged **then**
        **break**
      **end if**
    **end for**
    Regroup LoRA components with $\mathbf{\Pi}$
  **end for**

---

**Corollary C.4** (Necessary and sufficient condition for the regrouping invariance). *If gating coefficients $g_k(\mathbf{x})$ vary with $\mathbf{x}$ and the transfer/rescaling is fixed in the model definition, identical outputs for all $\mathbf{x}$ occur if and only if the ratio $g_u(\mathbf{x})/g_v(\mathbf{x})$ is constant (over all $\mathbf{x}$) for the transferred component. In that case, the constant defines $\gamma$, and the above proof applies to every $\mathbf{x}$ without change. If the ratio is input-dependent, the exact invariance rule can still be applied pointwise by setting $\gamma(\mathbf{x})$ according to $g_u(\mathbf{x})/g_v(\mathbf{x})$ for each forward computation.*

## D. Complete Algorithm

In Alg 1, we present the complete optimization procedure of our method BADIT.

## E. Computational Complexity

In this section, we give a formal computational complexity analysis of our proposed basic ability decomposition and dynamic regrouping.

### E.1. Complexity of Basic Ability Decomposition

We consider a single MLP layer weight matrix $\mathbf{W} \in \mathbb{R}^{m \times n}$, where $m$ is the output dimension and $n$ is the input dimension. Our goal is to decompose $\mathbf{W}$ into $K$ rank-$r$ LoRA experts. The specific process in question performs one

truncated SVD of $\mathbf{W}$, keeping the top $rK$ singular vectors/values, partitioning them into $K$ groups of $r$ vectors, one group per expert, and using the remaining singular vectors as the residual $\widehat{\mathbf{W}}$.

We often only need the top $rK$ singular components, where $rK \ll \min(m,n)$. Using algorithms such as Randomized SVD, we can compute only the first $rK$ singular values and vectors without forming the entire decomposition. In truncated SVD, the main operations consist of $rK$ iterations of matrix–vector multiplication, followed by orthogonalization. Both methods have a complexity of $\mathcal{O}(mnrK)$.

### E.2. Complexity of Dynamic Rank-1 Regrouping

Let $K$ be the number of experts, $r$ the LoRA rank per expert, and $m \times n$ the shape of each rank-1 component. The total number of components is $rK$. The dynamic regrouping consists of:

**Similarity computation.** We compute pairwise similarities between all $rK$ components: $\mathcal{O}(m+n)$. There are $r^2 K^2$ pairs, hence $\mathcal{T}_{\text{sim}} = \mathcal{O}(r^2 K^2 (m+n))$ and space $\mathcal{O}(r^2 K^2)$ to store the similarity matrix.

**Group assignment.** Given the similarity matrix, finding the optimal partition with capacity constraints can be done by greedy search: $\mathcal{O}(r^2 K^2)$ time. Space is dominated by the similarity matrix: $\mathcal{O}(r^2 K^2)$.

Therefore, the overall time complexity is $\mathcal{O}(r^2 K^2 (m+n))$, and space complexity is $\mathcal{O}(r^2 K^2)$.

## F. More Literature Reviews

In this section, we provide a comprehensive organization and overview of existing methods for multi-task instruct-tuning and parameter-efficient fine-tuning of LLMs.

### F.1. LLM Multi-Task Instruct-Tuning

According to recent surveys (Wang et al., 2024a; Shi et al., 2026), existing multi-task instruct-tuning approaches can be broadly categorized into three main paradigms: *rehearsal-based*, *optimization-based*, and *architecture-based* methods. Although multi-task learning has been extensively explored in the computer vision community, its application to LLMs remains in its early stages. Accordingly, this section provides a brief overview of general multi-task instruct-tuning techniques and their variants in LLMs.

**Rehearsal-based methods** directly store a portion of old data and re-train the model alongside new data when learning new tasks, thus preserving old knowledge (Lopez-Paz & Ranzato, 2017; Li et al., 2025). This approach is straightforward and effective, yet it encounters challenges such as storage costs (de Masson d'Autume et al., 2019; Huang

et al., 2024a). To tackle the issue of storage costs, several generative rehearsal-based methods employ generative models to generate data samples relevant to old tasks, thereby replacing the stored old task data. For example, they utilize variational auto-encoders (Shin et al., 2017) and generative adversarial networks (Liu et al., 2020; Wang et al., 2022a), to learn the data distribution of old tasks and generate data by sampling from this distribution in new tasks. On this basis, most existing efforts to address LLM cross-task interference involve synthesizing high-quality data for rehearsal using untrained proxy LLMs (Sun et al., 2020; Wang et al., 2024b). For example, SSR (Huang et al., 2024a) uses an untrained base model to generate task samples, and then employs the latest trained model to produce responses, thereby constructing rehearsal data pairs.

**Optimization-based methods** primarily focus on the gradients between tasks during model training, aiming to guarantee the optimization gradients across tasks do not interfere with each other (Wang et al., 2023a; Yuan et al., 2026). To achieve this, most existing approaches employ orthogonal optimization, aiming to make the gradients between tasks mutually orthogonal (Lopez-Paz & Ranzato, 2017; Saha et al., 2021) or to identify flat minima in the loss landscape during alternative task training (Li et al., 2024b; Bian et al., 2024). For example, O-LoRA (Wang et al., 2023a) learns distinct tasks in different low-rank subspaces while keeping orthogonality between these subspaces; GORP (Wang et al., 2025) restricts gradient updates to a low-rank subspace and ensures, via a projection operation, that this subspace is orthogonal to those used by previous tasks.

**Architecture-based methods** involve isolating parameters between tasks, and allocating a new set of learnable parameters for each new task's data (Wang et al., 2023b;a; Yang et al., 2025b) or extracting task-specific sub-networks from the pre-trained model (Chen et al., 2026). Subsequently, inference for new tasks is performed using techniques such as automatic routing (Dou et al., 2024; Ma et al., 2024; Feng et al., 2024). For example, LoRAMoE (Dou et al., 2024) designs a LoRA-based expert model for each task and employs a trained routing function for expert selection.

In this work, we also attempt to explain the mechanism of parameter overlap across different tasks, which has received relatively little attention. Song et al. (2024) identify the existence of task-specific neurons in LLMs and propose using special tokens to locate these neurons; Building on this, Leng & Xiong (2025) further demonstrate that the overlap among such task-specific parameters is strongly correlated with the cross-task generalization.

### F.2. Parameter-Efficient Fine-Tuning of LLMs

LLMs and their multimodal variants have demonstrated remarkable performance in diverse downstream tasks, e.g.,

mathematical reasoning (Liu et al., 2026; Huang et al., 2025). Nevertheless, their training costs remain prohibitively high (Liu et al., 2024a; Yan et al., 2026; Xie et al., 2026). Therefore, cutting-edge works propose *parameter-efficient fine-tuning* (PEFT) methods, with LoRA and MoE series receiving the most attention. Specifically, LoRA (Hu et al., 2022; Meng et al., 2024; Kou et al., 2026) is built on the assumption that the parameter changes during model adaptation to new tasks exhibit low-rank properties, and it possesses plug-and-play characteristics. For example, DoRA (Liu et al., 2024c) decomposes pretrained weights into magnitude and directional components; QLoRA (Dettmers et al., 2023) further quantizes LoRA, significantly reducing the fine-tuning costs of LLMs. Additionally, the MoE architecture (Shazeer et al., 2017; Gao et al., 2022), due to its sparsity and selective activation characteristics, has become the preferred choice for several open-source foundation LMs, e.g., Mixtral (Jiang et al., 2024) and DeepSeek (Liu et al., 2024a), and also serves as the base structure for numerous multi-task models (Ma et al., 2018; Huang et al., 2024b). More recently, partial works attempt to re-initialize LoRAs by select key dimensions in pre-trained weights through SVD (Meng et al., 2024; Yang et al., 2024; Fan et al., 2025), and this paper draws inspiration from these works to propose decomposing the basic abilities of LLMs.

# G. More Experimental Details

In this section, we provide a detailed description of our experimental setup, including dataset statistics, specifications of the LLMs used, baseline methods for comparison, implementation details, and formal evaluation metrics.

## G.1. Evaluation Benchmarks

*SuperNI* (Wang et al., 2022b) is a benchmark comprising a diverse set of NLP tasks paired with expert-written instructions, designed to enable rigorous evaluation of multi-task instruct-tuning methods for LLMs. Specifically, we follow Zhao et al. (2024b) to select three tasks each from five categories, including dialogue generation, information extraction, question answering, summarization, and sentiment analysis, resulting in a sequence of 15 tasks used to evaluate the various multi-task approaches. For each individual task, we truncate its training set to 1,000 samples and its validation set to 100 samples; for tasks with fewer samples than these thresholds, we retain their original sizes. Detailed statistics, domains, and evaluation metrics for these datasets are summarized in Table 4. Additionally, in Table 5 we show the task orderings corresponding to the sequential learning setting when the random seed is set to $\{1, 2, 3, 4, 5\}$, respectively.

## G.2. Model Cards

We employ three families of contemporary open-source: Qwen3 (Yang et al., 2025a), Llama3 (AI@Meta, 2024), Gemma2 (Rivière et al., 2024), represented by two parameter scales to evaluate performance across model sizes.

- **Qwen3-4B** [1] **/ 8B** [2] are dense-architecture LLMs, which are distilled variants derived via off-policy and on-policy knowledge distillation from larger teacher models, aiming to preserve strong reasoning and instruction-following capabilities while reducing computational cost.

- **Llama3-3B** [3] **/ 8B** [4] represents a significant upgrade over its predecessors with an expanded token vocabulary (128K), grouped-query attention, and training on over 15 trillion tokens.

- **Gemma2-2B** [5] **/ 9B** [6] are from a lightweight, open-weight model family built on the same research foundation as Gemini. It features innovations such as sliding-window attention, logit soft-capping, and extensive knowledge distillation.

## G.3. Baselines

We compare our approach against a representative set of parameter-efficient fine-tuning baselines, including two low-rank adaptation variants and three recent MoE-inspired methods tailored for multi-task instruct-tuning:

- **LoRA** (Hu et al., 2022) is a foundational PEFT method that freezes the pre-trained model weights and injects trainable low-rank decomposition matrices into attention layers, drastically reducing the number of trainable parameters while preserving performance close to full fine-tuning.

- **PiSSA** (Meng et al., 2024) improves upon LoRA by reinitializing the low-rank adapters using the principal components of the pre-trained weight matrix via SVD. Instead of random or zero initialization, PiSSA decomposes the original weight as $\mathbf{W} = \mathbf{U}\mathbf{\Sigma}\mathbf{V}^\top$ and initializes the adapter to capture the dominant singular directions, leading to faster convergence and higher downstream accuracy with the same parameter budget.

---

[1] https://huggingface.co/Qwen/Qwen3-4B
[2] https://huggingface.co/Qwen/Qwen3-8B
[3] https://huggingface.co/meta-llama/Llama-3.2-3B-Instruct
[4] https://huggingface.co/meta-llama/Meta-Llama-3-8B-Instruct
[5] https://huggingface.co/google/gemma-2-2b-it
[6] https://huggingface.co/google/gemma-2-9b-it

*Table 4.* Details of the multi-task datasets SuperNI (Wang et al., 2022b).

| Task ID | Dataset Name | Task | #Train | #Eval | Metrics |
|---------|-------------|------|--------|-------|---------|
| T002 | task002_quoref_answer_generation | question answering | 1,000 | 100 | ROUGE-L |
| T073 | task073_commonsenseqa_answer_generation | question answering | 975 | 100 | ROUGE-L |
| T591 | task591_sciq_answer_generation | question answering | 1,000 | 100 | ROUGE-L |
| T181 | task181_outcome_extraction | information extraction | 338 | 43 | ROUGE-L |
| T748 | task748_glucose_reverse_cause_event_detection | information extraction | 1,000 | 100 | ROUGE-L |
| T1510 | task1510_evalution_relation_extraction | information extraction | 1,000 | 100 | ROUGE-L |
| T363 | task363_sst2_polarity_classification | sentiment analysis | 1,000 | 100 | Accuracy |
| T875 | task875_emotion_classification | sentiment analysis | 1,000 | 100 | Accuracy |
| T1687 | task1687_sentiment140_classification | sentiment analysis | 1,000 | 100 | Accuracy |
| T511 | task511_reddit_tifu_long_text_summarization | text summarization | 1,000 | 100 | ROUGE-L |
| T1290 | task1290_xsum_summarization | text summarization | 1,000 | 100 | ROUGE-L |
| T1572 | task1572_samsum_summary | text summarization | 160 | 20 | ROUGE-L |
| T639 | task639_multi_woz_user_utterance_generation | dialogue generation | 142 | 18 | ROUGE-L |
| T1590 | task1590_diplomacy_text_generation | dialogue generation | 126 | 16 | ROUGE-L |
| T1729 | task1729_personachat_generate_next | dialogue generation | 1,000 | 100 | ROUGE-L |

*Table 5.* Five different random seeds correspond to five distinct task orderings under the sequential training setting.

| Seed | Task Ordering |
|------|--------------|
| 1 | T591 → T073 → T1687 → T875 → T1572 → T639 → T1510 → T1590 → T511 → T181 → T363 → T1729 → T002 → T1290 → T748 |
| 2 | T1290 → T1510 → T1572 → T002 → T073 → T511 → T1590 → T363 → T748 → T639 → T181 → T1729 → T1687 → T591 → T875 |
| 3 | T002 → T073 → T1572 → T181 → T591 → T1687 → T363 → T1729 → T511 → T875 → T639 → T748 → T1590 → T1290 → T1510 |
| 4 | T1290 → T875 → T639 → T1590 → T591 → T073 → T1687 → T002 → T748 → T511 → T1572 → T181 → T1729 → T363 → T1510 |
| 5 | T748 → T1510 → T875 → T002 → T591 → T1729 → T1572 → T511 → T1687 → T1590 → T073 → T639 → T181 → T363 → T1290 |

- **OLoRA** (Wang et al., 2023a) enhances LoRA's optimization stability and isolates different LoRA gradients by enforcing orthogonality in the low-rank adapter matrices. This structured initialization reduces gradient interference and accelerates training, particularly in low-data or multi-task regimes.

- **LoRAMoE** (Dou et al., 2024) integrates the MoE paradigm into LoRA by deploying multiple parallel LoRA adapters per layer, each acting as an expert, along with a trainable router that selects or blends experts based on input tokens. Crucially, it partitions experts into two groups, one dedicated to preserving world knowledge from pretraining and the other to learning task-specific instructions, thereby mitigating catastrophic forgetting during supervised fine-tuning.

- **GainLoRA** (Liang et al., 2025) addresses continual and multi-task learning by dynamically gating the contribution of task-specific LoRA branches during inference. It introduces a learnable gate module that modulates how much each LoRA adapter influences the output, with constraints applied during training to suppress interference from newly added adapters on previously learned tasks, thus improving knowledge retention across sequential instruct-tuning.

### G.4. Implementation Details

We implement our method based on the above 6 LLMs, using the Hugging Face Transformers library. All experiments are conducted on 8 NVIDIA 4090 GPUs with DeepSpeed ZeRO-2 enabled to support efficient distributed training without CPU offloading, and we enable FlashAttention-2 for accelerated attention computation. For parameter-efficient fine-tuning, we inject a MoE-style LoRA architecture into the gate projection of each transformer MLP block. The base LoRA configuration uses rank $r = 4$, scaling factor $\alpha = 32$ (yielding an effective scaling of $\alpha/r = 8$), and dropout rate of 0.05. Our method employs 8 experts, with a top-$k = 4$ routing strategy that activates four experts per token during training and inference. The expert selection and combination are learned end-to-end alongside the router parameters. Training is performed for 10 epochs on the *SuperNI* dataset, with a maximum sequence length of 1024 tokens and a target output length capped at 50 tokens. We use a per-device batch size of 1 and accumulate gradients over 1 step, resulting in an effective global batch size of 8. Optimization is carried out using the AdamW optimizer with a cosine learning rate scheduler, peak learning rate of $2 \times 10^{-4}$, warmup ratio of 0.03, and no weight decay. Mixed-precision training is enabled via bf16. We fix the random

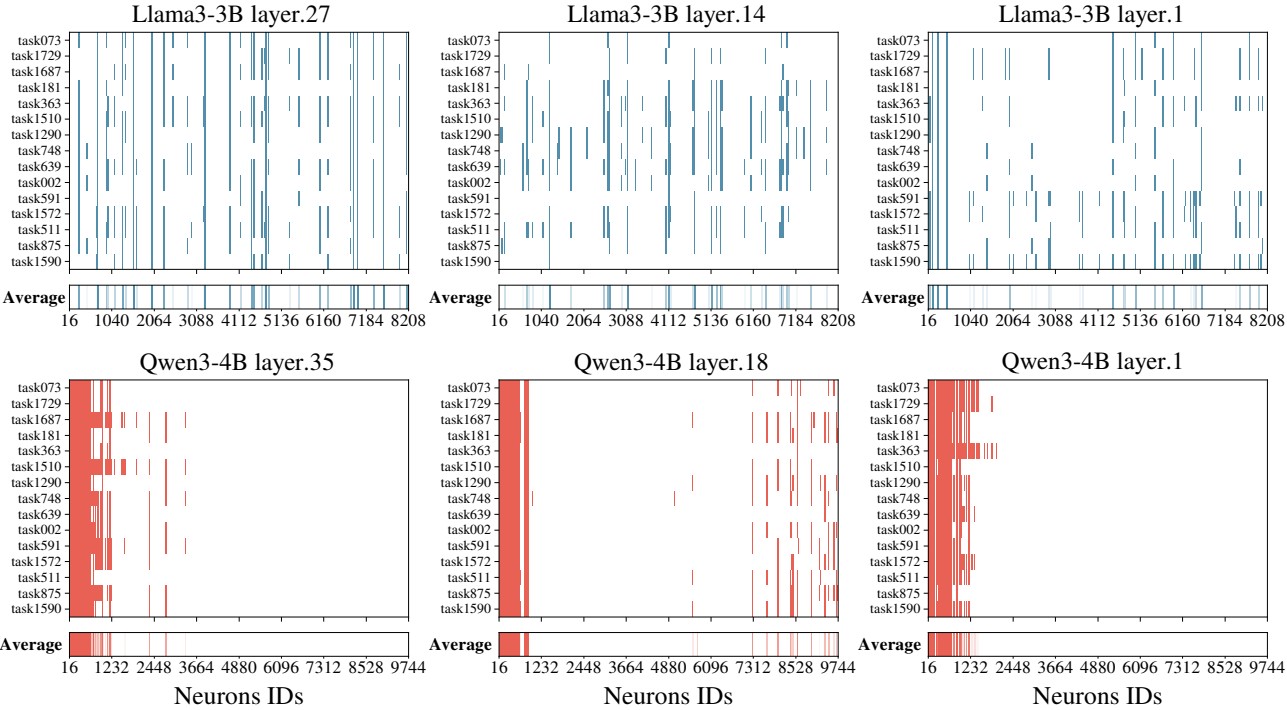

*Figure 5.* Shared task-specific neurons of MLP gate layers across different tasks.

seed to the values $\{1, 2, 3, 4, 5\}$ to ensure reproducibility.

### G.5. Evaluation Metrics

Following standard practice in previous literature (Lopez-Paz & Ranzato, 2017; Zhao et al., 2024b), we adopt the following four metrics to comprehensively assess model performance:

- **ROUGE** measures the overall effectiveness after learning the entire sequence of tasks as

$$\text{ROUGE} = \frac{1}{T} \sum_{t=1}^{T} a_{t,T},$$

  where denotes the model's performance on $t$-th task after training on the $T$-th task. This reflects the final model's ability to perform well across all seen tasks.

- **Forward** quantifies how much prior knowledge acquired from previously seen tasks helps (or hinders) the learning of a new task. It is measured relative to a baseline where each task is learned in isolation. Let $a_{t,0}$ denote the model's performance on the $t$-th task when trained alone (i.e., without any other tasks), Forward is defined as

$$\text{Forward} = \begin{cases} \frac{1}{T} \sum_{t=1}^{T} (a_{t,T} - a_{t,0}), & \text{mixed training,} \\ \frac{1}{T} \sum_{t=1}^{T} (a_{t,t} - a_{t,0}), & \text{sequential training.} \end{cases}$$

A positive Forward value indicates that leveraging prior experience improves performance compared to isolated training.

- **Forget Rate** quantifies the extent of catastrophic forgetting on previously learned tasks under the sequential training setting as

$$\text{Forget Rate} = \frac{1}{T-1} \sum_{t=1}^{T-1} (a_{t,t} - a_{t,T}).$$

A higher value indicates more severe forgetting, as it captures the performance drop from the peak (after learning $t$-th task) to the end (after learning all $T$ tasks).

- **Backward** measures the influence of learning subsequent tasks on the performance of earlier tasks under the sequential training setting as

$$\text{Backward} = \frac{1}{T-1} \sum_{t=1}^{T-1} (a_{t,T} - a_{t,t}).$$

Backward can be positive if later tasks improve earlier ones (positive backward transfer), making it a more general measure.

## H. More Experimental Results

In this section, we present additional experimental investigations and the results of our evaluation studies, including

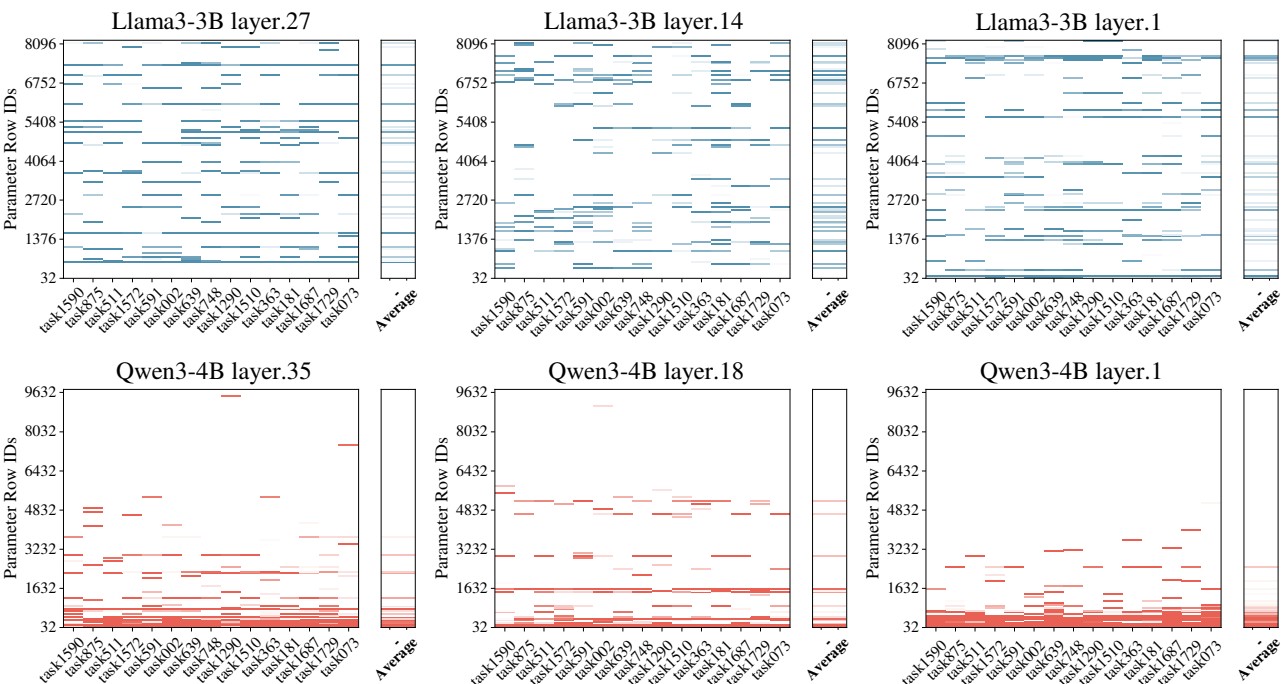

*Figure 6.* Shared task-specific parameters of MLP gate layers across different tasks.

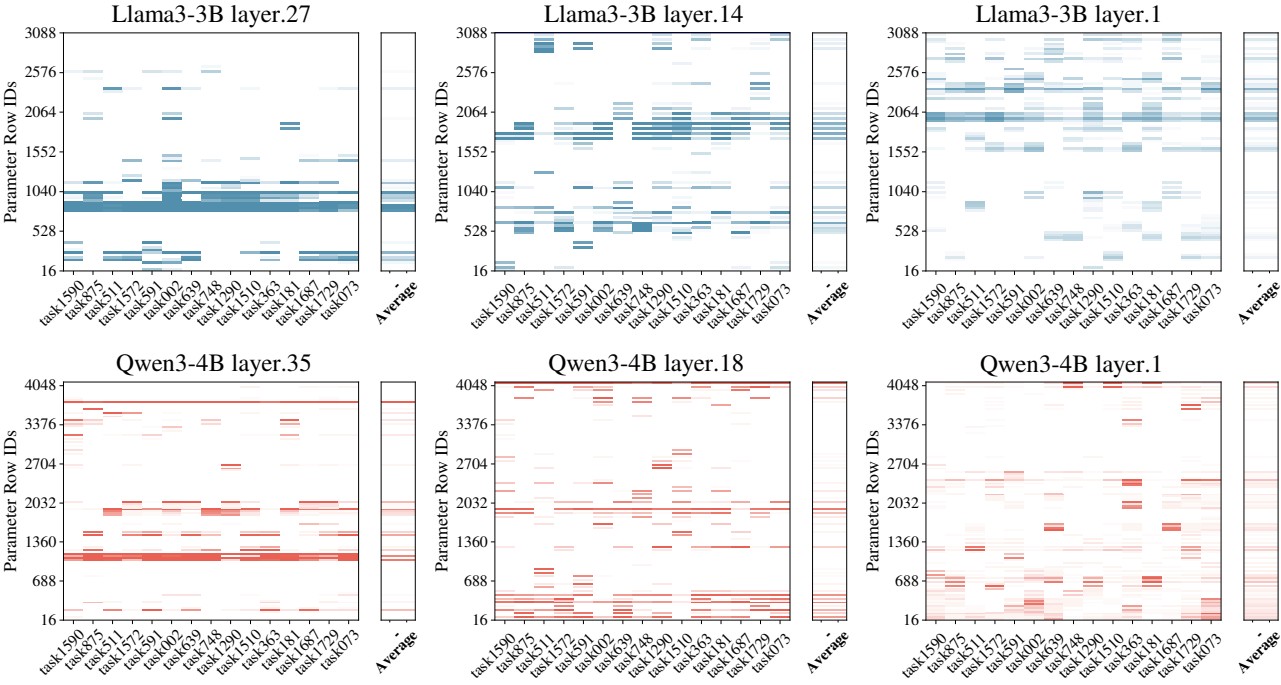

*Figure 7.* Shared task-specific parameters of query matrices in self-attention layers across different tasks.

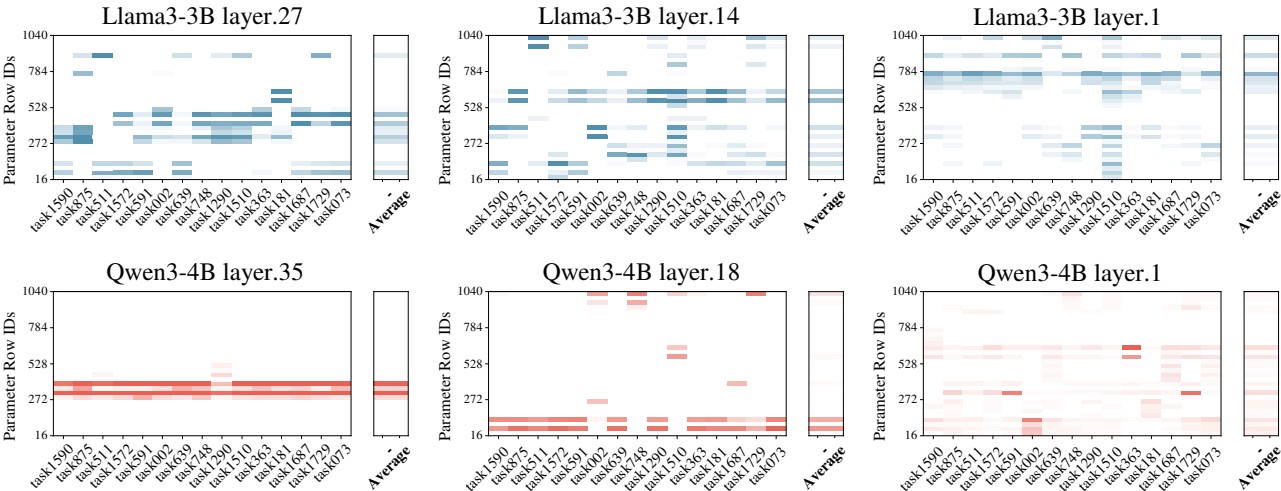

*Figure 8.* Shared task-specific parameters of key matrices in self-attention layers across different tasks.

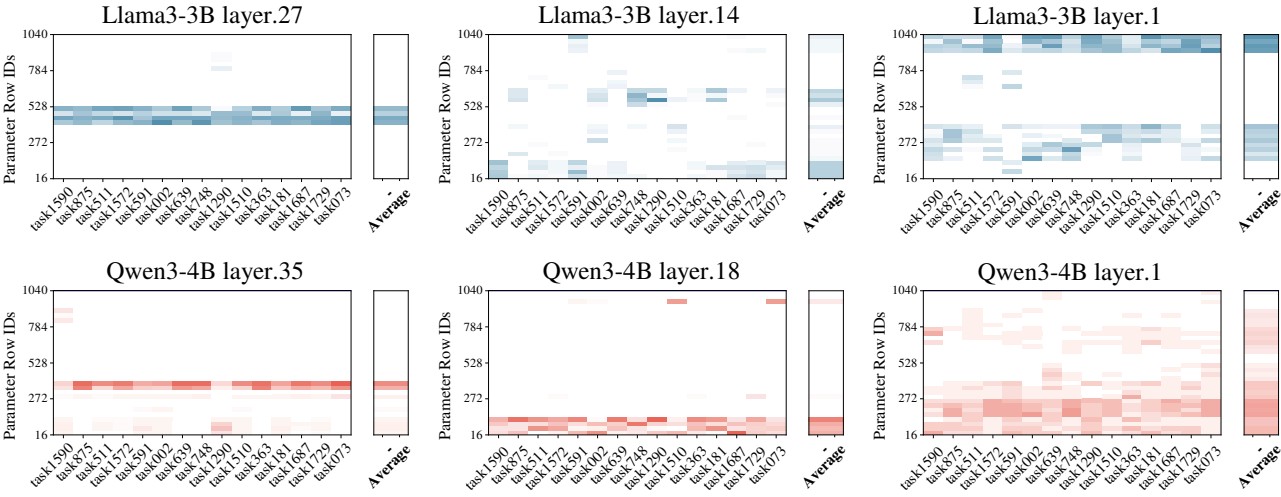

*Figure 9.* Shared task-specific parameters of value matrices in self-attention layers across different tasks.

gradient angles, sensitivity analysis, and performance across different tasks.

### H.1. More Investigation Results

In this section, we employ the same experimental setup as Sec. 2 to investigate neuron and parameter sharing behaviors across different tasks in a broader range of modules. Specifically, we conduct experiments on both Llama3-3B and Qwen3-4B and report their cross-task sharing performance in key components, including the gate projections in the MLP blocks and the query, key, and value matrices in the self-attention modules, in Figs. 5, 6, 7, 8, and 9. Furthermore, we examine these modules across Transformer layers at three distinct depths: low, middle, and high. For Llama3-3B, which comprises 28 layers in total, we select layers 1, 14, and 27 to represent the low, middle, and high

levels, respectively. Similarly, for Qwen3-4B, which has 36 layers, we analyze layers 1, 18, and 35 as representatives of the corresponding depth categories.

Overall, our findings are consistent with the conclusions drawn in Sec. 2: a substantial portion of neurons and parameters remains inactive across all tasks, while those that are co-activated continue to naturally organize into base compositional groups. Additionally, Qwen3-4B again exhibits the phenomenon where the majority of activated neurons are concentrated at lower neuron IDs.

*Comparing sharing behaviors across different layers*, we first observe that, for the gate projections in Figs. 5 and 6, activation patterns show no significant variation across layers, e.g., activations in Qwen3-4B consistently cluster at lower neuron IDs regardless of depth. In contrast, for the query, key, and value matrices in the self-attention modules,

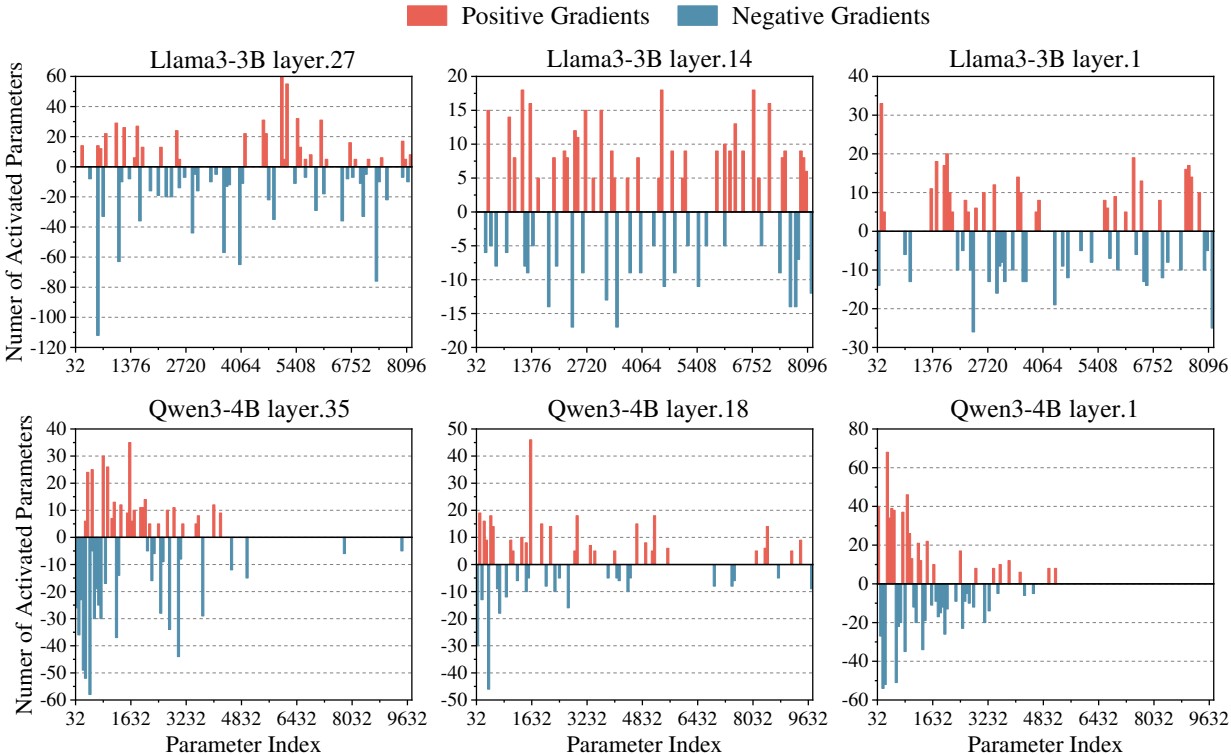

*Figure 10.* The number of parameters that activate each parameter column with positive or negative gradients across all tasks, respectively.

we identify systematic shifts in activation patterns from lower to higher layers. Specifically, activated parameters tend to migrate from higher to lower parameter IDs as depth increases, for instance, in the query matrix of Llama3-3B in Fig. 7, or transition from scattered, diffuse activations toward more concentrated clusters, as seen in the value matrix of Qwen3-4B in Fig. 9. Although these trends indicate that different layers exhibit distinct sharing behaviors, they all remain consistent with our core hypothesis: the parameter matrices in each layer can be decomposed into a set of basic, reusable abilities. Notably, prior work has suggested that only the gate projections require decoupling (Dai et al., 2022; Song et al., 2024). Given that the sharing patterns of gate matrices are largely consistent across layers in Fig. 6, applying the same decoupling strategy uniformly across all layers remains justified and effective.

Moreover, a potential concern arises from our primary focus on the magnitude of parameter activation, i.e., the absolute value of gradients, across tasks. Specifically, if different tasks induce gradients of opposite signs on the same parameter, they may exhibit conflicting optimization directions despite similar activation magnitudes. To investigate this, we present an additional experiment in Fig. 10, which visualizes, for each parameter, the number of tasks that produce positive versus negative gradients. In this figure, the vertical axis represents the total number of activated parameters

across 15 distinct tasks for each parameter row (i.e., the sum of tasks that activate that particular parameter, regardless of gradient sign). This analysis helps reveal whether conflicting gradient directions are prevalent and provides further insight into the compatibility of task-specific updates at the parameter level. As shown in Fig. 10, the majority of parameters are consistently activated with gradients of the same sign across different tasks; only a small fraction exhibit both positive and negative gradients simultaneously. This result demonstrates that, even when multiple tasks activate the same parameters, their gradient directions are largely aligned in most cases. Consequently, this further supports our central analogy: such shared activation patterns can be interpreted as the reuse of basic abilities across tasks.

### H.2. More Gradient Angles

In this section, we extend the intra-expert and inter-expert gradient angle results of our method BADIT, originally presented in Fig. 4, to a broader set of LLMs, with the extended results shown in Fig. 11. We conduct experiments across all 6 LLMs and evaluate two different task orders: *Task Order A* and *Task Order B*, which correspond to the training sequences using random seeds 1 and 2, respectively, as specified in Table 5. The experimental results align with the findings in Fig. 4: our method successfully maintains inter-expert gradient angles near 90 degrees, indicating ap-

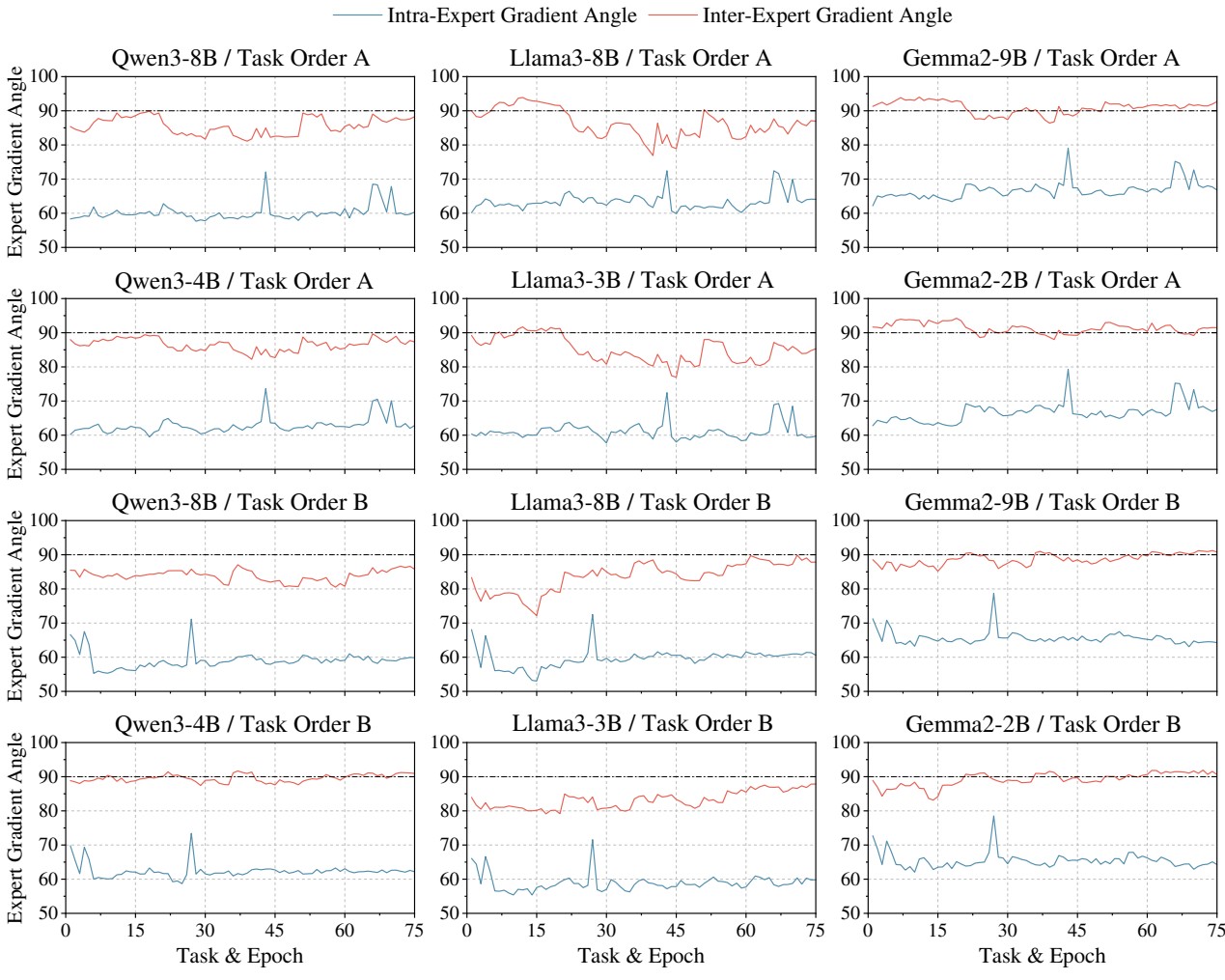

*Figure 11.* Intra- and inter-expert gradient angles of BADIT on 6 LLMs across epochs.

proximate orthogonality. In some cases, such as Qwen3-4B under Task Order B, the inter-expert angles remain consistently clustered around 90 degrees throughout training. Meanwhile, intra-expert gradient angles are consistently kept at relatively small values. These results collectively demonstrate the effectiveness of our DOG method in preserving orthogonality between experts while encouraging coherent updates within each expert.

### H.3. Sensitivity Analysis

To investigate the impact of the number of LoRA experts $K$ and the LoRA rank $r$ on model performance, we conduct a sensitivity analysis across three LLMs: Qwen3-8B, Llama3-8B, and Gemma2-9B, and present the results in Fig. 12. Generally, under both mixed training and sequential training settings, our method consistently achieves optimal performance when $K = 8$ and $r = 4$. Deviating from these values, either increasing or decreasing $K$ or $r$, leads to degraded performance.

In our method, $K$ corresponds to the number of basic abilities decoupled by BADIT. The empirical results indicate that decomposing the model into exactly 8 such abilities yields the best performance. When $K$ is too small, the set of decoupled abilities is insufficient to adequately represent all 15 tasks. Conversely, when $K$ is too large, redundant abilities emerge: this not only increases training overhead but also interferes with the routing mechanism by diluting its focus on the most relevant abilities. Similarly, the LoRA rank $r$ represents the granularity of each decoupled unit. If $r$ is too small, individual experts lack sufficient capacity to fully encode a distinct basic ability. On the other hand, an excessively large $r$, much like an oversized $K$, imposes greater computational costs during both training and the DOG process. Moreover, a larger $r$ introduces more candidate features during grouping, which complicates the clean separation of basic abilities and undermines the effectiveness of the grouping strategy.

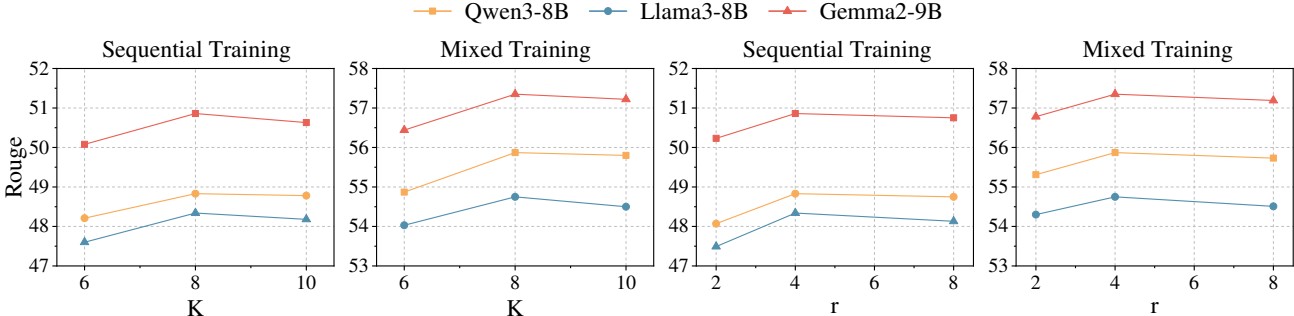

*Figure 12.* Sensitivity analysis of the parameters $K$ and $r$.

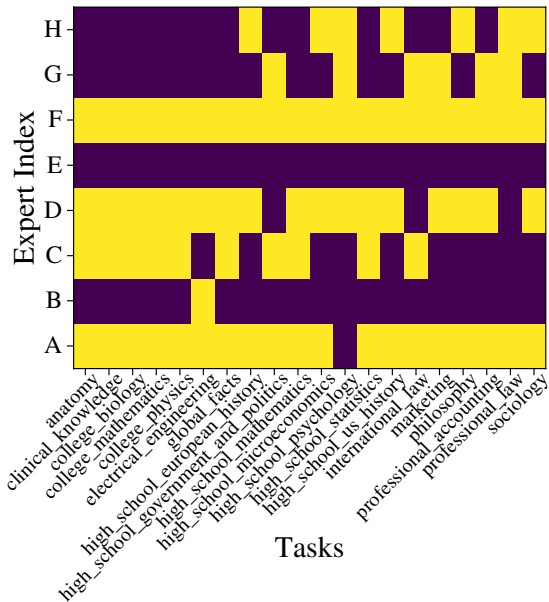

*Figure 13.* Visualizing basic abilities.

## H.4. Visualizing Basic Abilities

To further explore whether these decomposed LoRAs represent distinct skills, we conduct a toy experiment on *SuperNI* (K=8, top-4 experts activated) and evaluate expert activation on 20 *MMLU* tasks using our proposed BADIT. Their activated patterns are shown in Fig. 13.

Our qualitative analysis (where 1=activated, 0=inactive) reveals intriguing functional patterns: Experts A and F consistently activate across nearly all tasks, suggesting they encode general skills; Experts A, C, D, and F concurrently activate for factual knowledge and logical reasoning; Expert H exhibits a strong preference for history and sociology; Expert E is predominantly triggered by governance, business, and law-related tasks. These results suggest that expert activations follow identifiable functional patterns.

## H.5. Performance Across Different Tasks

We also provide detailed evaluation results under the sequential training setting, showing the performance of the LLMs on all 15 tasks after each new task is learned. We compare LoRAMoE against our BADIT method under two specific configurations: (1) training Qwen3-8B on the task sequence corresponding to random seed 1, and (2) training Gemma2-9B on the task sequence corresponding to random seed 3. The results for these experiments are presented in Tables 6, 7, 8, and 9, respectively.

The results clearly reveal the presence of catastrophic forgetting during sequential training: as the model learns new tasks, its performance on previously acquired tasks consistently degrades. For instance, Tasks 591 and 073, the first two tasks in the sequence, exhibit a steady decline in performance as training progresses. In this challenging setting, our method consistently outperforms the baseline LoRAMoE across all evaluation metrics. This not only demonstrates that our approach achieves superior overall multi-task performance but also provides strong evidence that it effectively mitigates task interference and alleviates catastrophic forgetting in sequential learning scenarios.

*Table 6.* Evaluation performance of the model across different tasks during sequential training, using the **Qwen3-8B** model with the **LoRAMoE** method (Rouge = 48.43, Forget Rate = 8.37, Backward = -5.71).

| | | Evaluation Task ID | | | | | | | | | | | | | |
|---|---|---|---|---|---|---|---|---|---|---|---|---|---|---|---|
| | | 591 | 073 | 1687 | 875 | 1572 | 639 | 1510 | 1590 | 511 | 181 | 363 | 1729 | 002 | 1290 | 748 |
| Training Task ID | 591 | 62.2 | 83.0 | 79.0 | 45.0 | 37.6 | 2.9 | 87.2 | 11.4 | 16.0 | 39.6 | 92.0 | 13.5 | 75.4 | 20.6 | 43.1 |
| | 073 | 68.1 | 82.7 | 79.0 | 43.0 | 38.1 | 4.5 | 79.5 | 13.6 | 15.3 | 27.6 | 90.0 | 14.4 | 75.7 | 19.5 | 41.6 |
| | 1687 | 69.3 | 78.0 | 81.0 | 37.0 | 33.3 | 3.6 | 73.3 | 12.6 | 16.0 | 23.8 | 91.0 | 16.0 | 77.9 | 19.7 | 42.4 |
| | 875 | 66.6 | 74.0 | 73.0 | 69.0 | 38.1 | 4.2 | 44.7 | 11.4 | 15.4 | 21.8 | 78.0 | 16.0 | 78.7 | 20.2 | 40.7 |
| | 1572 | 67.3 | 69.0 | 68.0 | 72.0 | 43.0 | 5.2 | 79.2 | 9.8 | 16.7 | 22.1 | 86.0 | 16.8 | 75.7 | 19.7 | 39.7 |
| | 639 | 60.6 | 70.0 | 63.0 | 67.0 | 36.5 | 12.3 | 77.8 | 11.7 | 15.8 | 16.6 | 77.0 | 13.9 | 76.7 | 19.5 | 39.8 |
| | 1510 | 62.9 | 71.0 | 78.0 | 69.0 | 43.1 | 15.6 | 100.0 | 11.8 | 14.8 | 36.5 | 86.0 | 16.6 | 77.8 | 20.2 | 43.0 |
| | 1590 | 58.4 | 69.0 | 72.0 | 66.0 | 41.4 | 12.4 | 100.0 | 11.8 | 15.7 | 39.5 | 84.0 | 15.2 | 75.5 | 20.6 | 44.0 |
| | 511 | 53.4 | 68.0 | 75.0 | 69.0 | 35.2 | 6.4 | 100.0 | 11.9 | 15.6 | 37.8 | 85.0 | 10.0 | 69.8 | 22.6 | 41.4 |
| | 181 | 51.3 | 64.0 | 74.0 | 68.0 | 32.9 | 8.0 | 99.7 | 11.0 | 12.8 | 69.2 | 71.0 | 13.3 | 73.3 | 21.9 | 42.2 |
| | 363 | 53.3 | 63.0 | 76.0 | 62.0 | 39.6 | 8.1 | 100.0 | 10.1 | 14.1 | 71.2 | 87.0 | 12.2 | 74.0 | 21.4 | 45.0 |
| | 1729 | 51.3 | 69.0 | 79.0 | 60.0 | 40.9 | 5.2 | 100.0 | 11.7 | 11.6 | 75.4 | 88.0 | 15.6 | 77.1 | 20.7 | 43.4 |
| | 002 | 52.1 | 61.0 | 82.0 | 56.0 | 39.7 | 5.1 | 100.0 | 11.6 | 12.8 | 62.4 | 88.0 | 14.0 | 77.2 | 21.9 | 42.0 |
| | 1290 | 46.6 | 59.0 | 75.0 | 55.0 | 18.2 | 8.1 | 99.0 | 7.3 | 8.4 | 68.0 | 85.0 | 12.3 | 71.1 | 25.7 | 34.2 |
| | 748 | 43.4 | 62.0 | 75.0 | 61.0 | 31.4 | 6.9 | 98.3 | 7.9 | 12.0 | 70.0 | 86.0 | 11.8 | 76.2 | 24.7 | 59.9 |

*Table 7.* Evaluation performance of the model across different tasks during sequential training, using the **Qwen3-8B** model with the **BADIT** method (Rouge = 49.96, Forget Rate = 7.27, Backward = -5.35).

| | | Evaluation Task ID | | | | | | | | | | | | | |
|---|---|---|---|---|---|---|---|---|---|---|---|---|---|---|---|---|
| | | 591 | 073 | 1687 | 875 | 1572 | 639 | 1510 | 1590 | 511 | 181 | 363 | 1729 | 002 | 1290 | 748 |
| Training Task ID | 591 | 69.2 | 78.0 | 80.0 | 44.0 | 41.8 | 5.1 | 90.8 | 14.1 | 16.3 | 50.7 | 91.0 | 13.7 | 77.7 | 20.1 | 41.7 |
| | 073 | 70.6 | 79.5 | 76.0 | 48.0 | 38.9 | 5.5 | 96.5 | 11.4 | 15.7 | 40.8 | 91.0 | 17.2 | 80.5 | 21.3 | 43.0 |
| | 1687 | 69.4 | 77.0 | 86.0 | 43.0 | 37.1 | 6.7 | 91.7 | 11.6 | 16.3 | 33.8 | 87.0 | 15.8 | 81.3 | 20.6 | 40.1 |
| | 875 | 69.7 | 76.0 | 83.0 | 73.0 | 39.7 | 3.6 | 47.4 | 11.1 | 17.7 | 29.2 | 81.0 | 14.6 | 72.7 | 19.7 | 38.4 |
| | 1572 | 67.9 | 75.0 | 81.0 | 75.0 | 39.7 | 6.3 | 76.8 | 12.0 | 15.9 | 30.9 | 79.0 | 13.6 | 80.4 | 20.6 | 43.3 |
| | 639 | 61.1 | 71.0 | 82.0 | 75.0 | 37.4 | 12.8 | 63.8 | 14.3 | 14.3 | 30.2 | 81.0 | 14.0 | 73.5 | 20.3 | 41.0 |
| | 1510 | 65.9 | 74.0 | 81.0 | 76.0 | 39.8 | 11.7 | 100.0 | 10.8 | 15.4 | 34.5 | 83.0 | 14.1 | 80.3 | 19.3 | 42.2 |
| | 1590 | 63.4 | 72.0 | 83.0 | 74.0 | 35.8 | 7.9 | 99.7 | 11.4 | 15.0 | 34.3 | 82.0 | 11.1 | 76.9 | 19.5 | 42.7 |
| | 511 | 56.6 | 75.0 | 83.0 | 73.0 | 32.5 | 8.9 | 100.0 | 8.6 | 15.3 | 29.2 | 84.0 | 10.5 | 71.9 | 22.2 | 37.3 |
| | 181 | 55.8 | 75.0 | 82.0 | 73.0 | 28.5 | 6.9 | 100.0 | 10.3 | 15.5 | 70.8 | 83.0 | 11.0 | 76.1 | 21.6 | 41.9 |
| | 363 | 60.6 | 73.0 | 81.0 | 70.0 | 33.9 | 9.7 | 100.0 | 10.3 | 15.8 | 71.1 | 91.0 | 11.1 | 76.6 | 21.3 | 43.9 |
| | 1729 | 60.4 | 72.0 | 80.0 | 68.0 | 34.7 | 10.7 | 99.7 | 11.2 | 15.1 | 65.9 | 92.0 | 12.8 | 77.7 | 21.1 | 44.2 |
| | 002 | 54.6 | 74.0 | 80.0 | 62.0 | 35.8 | 11.0 | 100.0 | 11.4 | 15.3 | 64.1 | 90.0 | 15.1 | 77.2 | 19.7 | 40.8 |
| | 1290 | 52.7 | 76.0 | 79.0 | 58.0 | 27.5 | 8.4 | 100.0 | 9.7 | 10.6 | 60.6 | 84.0 | 12.6 | 77.9 | 26.0 | 28.9 |
| | 748 | 53.5 | 75.0 | 81.0 | 56.0 | 29.1 | 7.6 | 99.0 | 11.8 | 12.7 | 55.3 | 89.0 | 13.6 | 76.4 | 24.4 | 65.1 |

*Table 8.* Evaluation performance of the model across different tasks during sequential training, using the **Gemma2-9B** model with the **LoRAMoE** method (Rouge = 47.05, Forget Rate = 11.94, Backward = -10.11).

| | | Evaluation Task ID | | | | | | | | | | | | | | |
|---|---|---|---|---|---|---|---|---|---|---|---|---|---|---|---|---|
| | | *002* | *073* | *1572* | *181* | *591* | *1687* | *363* | *1729* | *511* | *875* | *639* | *748* | *1590* | *1290* | *1510* |
| Training Task ID | *002* | 83.1 | 49.0 | 34.6 | 26.4 | 67.7 | 73.0 | 87.2 | 12.6 | 15.6 | 45.0 | 4.7 | 25.4 | 7.5 | 21.2 | 89.0 |
| | *073* | 83.2 | 66.0 | 37.1 | 32.4 | 71.4 | 73.0 | 82.0 | 12.0 | 15.1 | 43.0 | 3.6 | 22.7 | 5.4 | 21.1 | 84.8 |
| | *1572* | 79.8 | 76.0 | 30.0 | 65.7 | 66.8 | 75.0 | 93.0 | 15.5 | 14.9 | 46.0 | 5.6 | 39.9 | 11.7 | 21.1 | 97.0 |
| | *181* | 83.2 | 83.0 | 33.3 | 47.0 | 63.5 | 75.0 | 94.0 | 12.1 | 14.7 | 44.0 | 3.7 | 27.7 | 9.5 | 20.7 | 96.0 |
| | *591* | 81.7 | 77.0 | 41.0 | 50.4 | 63.9 | 72.7 | 92.0 | 14.9 | 14.9 | 46.0 | 8.2 | 41.2 | 12.4 | 21.4 | 92.9 |
| | *1687* | 81.5 | 82.0 | 38.3 | 43.9 | 63.9 | 72.0 | 92.0 | 12.9 | 16.1 | 47.0 | 8.3 | 29.7 | 13.8 | 21.6 | 95.1 |
| | *363* | 81.7 | 78.0 | 40.2 | 77.4 | 61.6 | 74.0 | 86.0 | 14.2 | 16.4 | 41.0 | 3.7 | 28.1 | 12.1 | 20.3 | 94.8 |
| | *1729* | 82.8 | 81.0 | 41.1 | 73.5 | 69.6 | 68.0 | 90.0 | 11.8 | 15.3 | 42.0 | 3.7 | 22.9 | 10.0 | 20.7 | 92.5 |
| | *511* | 83.4 | 69.0 | 37.5 | 70.5 | 69.8 | 70.0 | 87.0 | 8.2 | 14.4 | 40.0 | 1.2 | 21.4 | 4.2 | 18.9 | 83.3 |
| | *875* | 78.7 | 36.0 | 32.9 | 72.7 | 72.3 | 72.0 | 90.0 | 7.6 | 13.8 | 45.0 | 0.8 | 19.6 | 2.6 | 18.6 | 77.0 |
| | *639* | 84.4 | 69.0 | 37.2 | 66.8 | 71.0 | 86.0 | 81.0 | 8.9 | 12.4 | 46.0 | 1.1 | 19.9 | 13.7 | 20.7 | 88.4 |
| | *748* | 79.9 | 43.0 | 30.6 | 65.7 | 71.9 | 87.0 | 88.0 | 6.9 | 13.4 | 45.0 | 0.8 | 18.8 | 2.9 | 19.2 | 83.3 |
| | *1590* | 81.7 | 58.0 | 40.3 | 57.2 | 72.1 | 78.0 | 92.0 | 12.8 | 12.7 | 35.0 | 0.4 | 16.1 | 11.5 | 18.9 | 80.3 |
| | *1290* | 77.9 | 62.0 | 34.7 | 56.5 | 74.4 | 80.0 | 89.0 | 11.1 | 11.7 | 40.0 | 1.6 | 20.0 | 11.8 | 19.8 | 72.5 |
| | *1510* | 79.9 | 57.0 | 38.5 | 62.4 | 70.5 | 77.0 | 91.0 | 17.4 | 13.4 | 35.0 | 6.4 | 18.7 | 10.4 | 19.4 | 91.4 |

*Table 9.* Evaluation performance of the model across different tasks during sequential training, using the **Gemma2-9B** model with the **BADIT** method (Rouge = 50.23, Forget Rate = 7.72, Backward = -6.39).

| | | Evaluation Task ID | | | | | | | | | | | | | | |
|---|---|---|---|---|---|---|---|---|---|---|---|---|---|---|---|---|
| | | *002* | *073* | *1572* | *181* | *591* | *1687* | *363* | *1729* | *511* | *875* | *639* | *748* | *1590* | *1290* | *1510* |
| Training Task ID | *002* | 82.5 | 70.0 | 34.0 | 28.3 | 71.6 | 73.0 | 89.0 | 12.2 | 15.7 | 43.0 | 4.5 | 30.1 | 8.0 | 21.8 | 93.0 |
| | *073* | 82.7 | 81.0 | 30.8 | 53.3 | 68.0 | 74.0 | 92.0 | 14.6 | 15.4 | 45.0 | 6.9 | 41.0 | 8.7 | 21.0 | 97.3 |
| | *1572* | 82.3 | 78.0 | 40.5 | 45.0 | 70.0 | 73.0 | 91.0 | 13.9 | 15.0 | 46.0 | 9.4 | 34.9 | 11.9 | 21.8 | 96.1 |
| | *181* | 76.0 | 77.0 | 39.8 | 68.4 | 66.2 | 74.0 | 90.0 | 13.4 | 15.3 | 40.0 | 5.4 | 30.0 | 13.5 | 20.7 | 87.8 |
| | *591* | 77.7 | 73.0 | 36.0 | 64.1 | 72.5 | 69.0 | 90.0 | 8.3 | 13.8 | 41.0 | 2.6 | 22.1 | 4.2 | 19.2 | 83.8 |
| | *1687* | 76.5 | 68.0 | 41.0 | 62.2 | 72.8 | 85.0 | 90.0 | 8.0 | 11.3 | 42.0 | 2.4 | 16.6 | 4.9 | 18.1 | 85.8 |
| | *363* | 78.3 | 77.0 | 37.5 | 62.6 | 70.7 | 81.0 | 92.0 | 10.6 | 9.5 | 44.0 | 2.2 | 15.6 | 11.5 | 19.2 | 80.3 |
| | *1729* | 77.5 | 75.0 | 40.7 | 69.5 | 69.0 | 80.0 | 94.0 | 14.6 | 12.4 | 39.0 | 6.5 | 16.7 | 11.1 | 20.2 | 88.2 |
| | *511* | 77.4 | 58.0 | 25.7 | 68.6 | 71.5 | 81.0 | 92.0 | 14.0 | 16.9 | 38.0 | 3.2 | 26.1 | 12.2 | 21.8 | 78.7 |
| | *875* | 76.1 | 45.0 | 31.2 | 66.7 | 68.2 | 60.0 | 78.0 | 14.5 | 16.0 | 84.0 | 1.3 | 14.8 | 9.9 | 21.2 | 50.4 |
| | *639* | 76.6 | 54.0 | 19.5 | 63.4 | 63.8 | 58.0 | 78.0 | 14.3 | 15.9 | 84.0 | 9.1 | 19.4 | 12.1 | 21.1 | 61.7 |
| | *748* | 75.9 | 53.0 | 36.7 | 71.2 | 63.3 | 77.0 | 79.0 | 16.7 | 16.6 | 85.0 | 10.4 | 62.8 | 12.5 | 19.6 | 59.6 |
| | *1590* | 74.6 | 53.0 | 34.2 | 71.5 | 63.6 | 72.0 | 80.0 | 15.0 | 16.6 | 82.0 | 10.6 | 63.3 | 12.6 | 21.5 | 60.3 |
| | *1290* | 73.7 | 64.0 | 17.3 | 61.0 | 61.5 | 58.5 | 84.0 | 13.3 | 11.2 | 79.0 | 7.7 | 61.2 | 11.2 | 27.6 | 55.6 |
| | *1510* | 75.7 | 68.0 | 15.5 | 64.1 | 60.9 | 75.0 | 83.0 | 13.6 | 11.1 | 82.0 | 4.9 | 62.6 | 11.2 | 26.0 | 100.0 |

