# OpenReview forum: "Decomposing the Basic Abilities of Large Language Models: Mitigating Cross-Task Interference in Multi-Task Instruct-Tuning"
_ICML.cc/2026/Conference — ICML 2026 regular_

### Official Review · Reviewer_B4w5 · 2026-03-06

**Soundness:** 3
**Presentation:** 4
**Significance:** 2
**Originality:** 3
**Overall Recommendation:** 4
**Confidence:** 4

**Summary:**

This paper tackles the problem of cross-task interference during the multi-task instruct-tuning of large language models (LLMs). The authors observe that existing methods, such as Mixture-of-Experts (MoE), fail to fully isolate parameters. To solve this, they propose BADIT to represent the "basis ability" and employs spherical clustering to dynamically maintain gradient orthogonality during training (DOG stage). Experimental on several superNI tasks demonstrate the effectiveness of methods.

**Compliance With Llm Reviewing Policy:**

Affirmed.

**Final Justification:**

I think most of my questions are addressed. So I raise my score.

**Key Questions For Authors:**

1. Why spherical clustering? Maybe you need more details to explain it.
2. Why rank-1 components within LoRA?
3. line 864: "Crucially, it partitions experts into two groups, one dedicated to preserving world knowledge from pretraining and the other to learning task-specific instructions", correct me if I am wrong; I do not know that LoRAMoE partitions experts into two groups.
4. I am not sure the random seed "1,2,3,4,5" is a good choice.

**Limitations:**

yes

**Strengths And Weaknesses:**

# Strength

1. the paper is well-written and well-motivated.
2. The authors successfully demonstrate that task-specific parameters are frequently shared and co-activated across up to 15 different tasks, providing a solid rationale for decomposing the model into orthogonal "basic abilities".
3. The combination of SVD-based initialization with dynamic gradient clustering is interesting.
4. The experimental setup is rigorous. The authors test their method on diverse LLMs and evaluate under both mixed and sequential training paradigms.
5. BADIT quantitatively outperforms strong, recent baselines like GainLoRA and PiSSA.
# weakness:

1. The key limitation of the paper is the Rouge score evaluation. The improvement is really limited compared to other baselines.
2. The paper claims to decompose "basic abilities," but relies entirely on mathematical orthogonality to define these abilities. There is no qualitative analysis exploring whether these experts map to interpretable, human-understandable skills (e.g., logic, sentiment, summarization).
3. The dynamic regrouping (DOG) stage relies on spherical K-means clustering and integer optimization, which are computationally intensive CPU operations.

---

> ### Author Rebuttal · Authors · 2026-03-29
>
> In response to the points raised, we provide the following explanations.
>
> > W1. The key limitation of the paper is the ROUGE score evaluation.
>
> Thank you. Our experiments strictly follow community standards. Unlike prior studies that utilized non-instruct-tuned LLMs (e.g., T5) as baselines, we evaluate on SOTA LLMs (e.g., Qwen3), which already possess formidable instruct-following capabilities. Therefore, achieving a ~2% improvement over such strong SOTA LLMs is highly non-trivial and represents a significant advancement. Additionally, the majority of our selected evaluation tasks involve highly structured outputs, e.g., multiple-choice selection. In these tasks, ROUGE serves as a highly reliable proxy for exact-match accuracy.
>
> > W2. There is no qualitative analysis exploring whether these experts map to interpretable, human-understandable skills.
>
> Thank you. Our motivation is that multi-task instruct-tuning inevitably suffers from parameter interference. Inspired by the co-activation, we hypothesize that different tasks inherently share underlying basic abilities. Accordingly, we decouple these abilities and maintain their orthogonality to enhance both learning efficiency. To further explore whether these decomposed LoRAs represent distinct skills, we conduct a toy experiment that evaluates expert activation on 20 MMLU tasks by BADIT. Their activated patterns are as follows:
>
> Task|A|B|C|D|E|F|G|H
> -|-|-|-|-|-|-|-|-
> anatomy|1|0|1|1|0|1|0|0
> clinical_knowledge|1|0|1|1|0|1|0|0
> college_biology|1|0|1|1|0|1|0|0
> college_mathematics|1|0|1|1|0|1|0|0
> college_physics|1|0|1|1|0|1|0|0
> electrical_engineering|1|1|0|1|0|1|0|0
> global_facts|1|0|1|1|0|1|0|0
> high_school_european_history|1|0|0|1|0|1|0|1
> high_school_government_and_politics|1|0|1|0|0|1|1|0
> high_school_mathematics|1|0|1|1|0|1|0|0
> high_school_microeconomics|1|0|0|1|0|1|0|1
> high_school_psychology|0|0|0|1|0|1|1|1
> high_school_statistics|1|0|1|1|0|1|0|0
> high_school_us_history|1|0|0|1|0|1|0|1
> international_law|1|0|1|0|0|1|1|0
> marketing|1|0|0|1|0|1|1|0
> philosophy|1|0|0|1|0|1|0|1
> professional_accounting|1|0|0|1|0|1|1|0
> professional_law|1|0|0|0|0|1|1|1
> sociology|1|0|0|1|0|1|0|1
>
> Our qualitative analysis (1=activated, 0=inactive) reveals: Experts 1 and 6 consistently activate across nearly all tasks, suggesting they encode general skills; Experts 1, 3, 4, and 6 concurrently activate for factual and logical knowledge; Expert 8 exhibits a strong preference for history and sociology; Expert 7 is always triggered by governance and law-related tasks. These results suggest that expert activations follow identifiable functional patterns. In future work, we plan to explore explicit constraints (e.g., constrained routing) to further enforce the disentanglement of these abilities.
>
> > W3. The DOG stage relies on spherical K-means clustering and integer optimization, which are computationally intensive CPU operations.
>
> Thank you. While the training time is approximately 22% higher than LoRAMoE (Table 3), our method maintains identical real-world inference time, due to the same model structure and parameter scale. Meanwhile, unlike prior MoE methods that scale expert counts linearly with new tasks, our focus on orthogonal basic abilities avoids this bottleneck. From the empirical performance, given the ~2% performance gain and the consistent mitigation of the forget rate, without any added inference latency, we believe this training overhead is well-justified.
>
> > Q1. Why spherical clustering?
>
> Thank you. The DOG stage aims to cluster rank-1 LoRA components with similar gradient directions. Because **gradient directions are inherently formulated as unit vectors**, spherical clustering is mathematically more appropriate than other clustering algorithms, as it naturally accounts for the directional geometry of the vector space.
>
> > Q2. Why rank-1 components within LoRA?
>
> Thank you. Our core idea involves regrouping the $rK$ high-singular-value eigenvectors from SVD so that each group represents a distinct basic ability. By utilizing the decomposition in Eq.(6), **each rank-1 LoRA component directly corresponds to an SVD eigenvector**. Therefore, regrouping these rank-1 components allows us to dynamically isolate and define specific abilities.
>
> > Q3. I do not know that LoRAMoE partitions experts into two groups.
>
> Thank you. LoRAMoE freezes the LLM's pre-trained weights to retain general knowledge while injecting LoRAs to capture task-specific knowledge. We acknowledge that our phrase *dividing experts into two groups* was imprecise; it is more accurate to state that it *partitions parameters into two groups.* We will correct it in the revision.
>
> > Q4. I am not sure the random seed "1,2,3,4,5" is a good choice.
>
> Thank you. We use the random seeds strictly to control the task orders during sequential training (Table 5). Unlike prior works that rely on a fixed task sequence, averaging results across multiple random task orders ensures the statistical reliability and stability of experiments.

---

> > ### Author Rebuttal · Reviewer_B4w5 · 2026-04-03
> >
> > Thanks for your response. I think most of my questions are addressed. I will raise my score.

---

> > > ### Author Response · Authors · 2026-04-04
> > >
> > > Thanks once again for your valuable comments. We are delighted to hear that your concerns have been resolved, and we are very grateful for your positive evaluation of our work.

---

### Official Review · Reviewer_Tt42 · 2026-03-10

**Soundness:** 3
**Presentation:** 3
**Significance:** 3
**Originality:** 3
**Overall Recommendation:** 4
**Confidence:** 3

**Summary:**

BADIT addresses the cross-task interference issue in multi-task instruction fine-tuning for LLMs. It proposes to decompose the model parameters into orthogonal "basic capability" experts (the BAD step), and dynamically maintain the orthogonality among experts during training through spherical clustering (the DOG step). Six LLMs were tested on the SuperNI benchmark, and the results were overall superior to the existing SOTA methods.

**Compliance With Llm Reviewing Policy:**

Affirmed.

**Final Justification:**

The rebuttals have resolved my questions and I will keep the positive score.

**Key Questions For Authors:**

see weekness.

**Limitations:**

Yes

**Strengths And Weaknesses:**

Advantages

The approach has a novel perspective: shifting from "isolated tasks" to "isolated capability units", which is more fundamental in logic.
The experimental design is solid: 6 models, dual training paradigms, complete ablation studies, and 5 random seed repetitions.
The visual verification of orthogonality (Fig. 4/11) is intuitive and persuasive.
The computational cost is reasonable, approximately 1.22 times the LoRAMoE training time.

Disadvantages
1. The core assumption lacks direct proof.
The paper jumps directly from "observing the co-activation phenomenon" to "the existence of orthogonal basic capabilities in LLM", and this inference is logically leap-like. Co-activation could very well be just a statistical characteristic of the parameter matrix (such as the by-product of low-rank structure), and does not necessarily correspond to meaningful "capability units". The paper does not provide any path for refutation, making this core concept more like a narrative framework rather than a verified assumption.
2. Completely lacks explainability experiments.
If "basic capabilities" truly exist, the most direct way to verify it is to show what each LoRA expert has learned - for example, one expert specializes in sentiment analysis, and another in abstract generation. The paper completely fails to conduct such analyses. The core selling point of "basic capabilities" thus remains at a metaphorical level and cannot be distinguished from the orthogonal LoRA decomposition in engineering.
3. Performance improvement figures have selective presentation.
The paper claims an average improvement of 2.68 ROUGE points, but this is the result of a comprehensive average across 6 models and both mixed and sequential training settings. When broken down, the improvement in single models and single settings is mostly between 1 and 2 points, and some models (such as Llama3-3B mixed training: 51.45 vs 49.72) have limited improvement. ROUGE itself has a coarse granularity, and the practical value of this magnitude of difference is also questionable.

---

> ### Author Rebuttal · Authors · 2026-03-29
>
> We appreciate your thorough and constructive feedback. In response to the points raised, we provide the following explanations.
>
> > **W1.** The core assumption lacks direct proof. The paper jumps directly from "observing the co-activation phenomenon" to "the existence of orthogonal basic capabilities in LLM", and this inference is logically leap-like.
>
> Thank you. Our preliminary experiments reveal that existing multi-task instruct-tuning methods suffer from significant overlap between task-specific parameters, causing inter-task interference. Inspired by the empirically observed co-activation, we hypothesize that different tasks inherently share a set of underlying *basic abilities*. Accordingly, to find these abilities, we decouple these abilities with SVD and maintain their orthogonality to enhance both learning efficiency and performance. Therefore, by this decomposition, our approach shifts the focus from the overlap of expert parameters across tasks of prior MoE-based methods to the sharing of orthogonal basic abilities across tasks, effectively mitigating multi-task interference.
>
> > **W2.** Completely lacks explainability experiments. If "basic capabilities" truly exist, the most direct way to verify it is to show what each LoRA expert has learned.
>
> Thank you. Our central claim is that multi-task instruction-tuning inevitably suffers from parameter interference. Inspired by the co-activation phenomenon in Fig.1, we hypothesize that different tasks inherently share a set of underlying basic abilities. Accordingly, we decouple these abilities and maintain their orthogonality to enhance both learning efficiency and overall performance. To further explore whether these decomposed LoRAs represent distinct skills, we conduct a toy experiment on SuperNI (K=8, top-4 experts activated) and evaluate expert activation on 20 MMLU tasks by our proposed BADIT. Their activated patterns are as follows:
>
> Task|A|B|C|D|E|F|G|H
> -|-|-|-|-|-|-|-|-
> anatomy|1|0|1|1|0|1|0|0
> clinical_knowledge|1|0|1|1|0|1|0|0
> college_biology|1|0|1|1|0|1|0|0
> college_mathematics|1|0|1|1|0|1|0|0
> college_physics|1|0|1|1|0|1|0|0
> electrical_engineering|1|1|0|1|0|1|0|0
> global_facts|1|0|1|1|0|1|0|0
> high_school_european_history|1|0|0|1|0|1|0|1
> high_school_government_and_politics|1|0|1|0|0|1|1|0
> high_school_mathematics|1|0|1|1|0|1|0|0
> high_school_microeconomics|1|0|0|1|0|1|0|1
> high_school_psychology|0|0|0|1|0|1|1|1
> high_school_statistics|1|0|1|1|0|1|0|0
> high_school_us_history|1|0|0|1|0|1|0|1
> international_law|1|0|1|0|0|1|1|0
> marketing|1|0|0|1|0|1|1|0
> philosophy|1|0|0|1|0|1|0|1
> professional_accounting|1|0|0|1|0|1|1|0
> professional_law|1|0|0|0|0|1|1|1
> sociology|1|0|0|1|0|1|0|1
>
> Our qualitative analysis (where 1=activated, 0=inactive) reveals intriguing functional patterns: Experts 1 and 6 consistently activate across nearly all tasks, suggesting they encode general skills; Experts 1, 3, 4, and 6 concurrently activate for factual knowledge and logical reasoning; Expert 8 exhibits a strong preference for history and sociology; Expert 7 is predominantly triggered by governance, business, and law-related tasks. These results suggest that expert activations follow identifiable functional patterns. We will include an exhaustive analysis across additional tasks and LLMs in the Appendix of the revised version. In future work, we plan to explore explicit constraints (e.g., constrained routing) to further enforce the disentanglement of these abilities.
>
> > **W3.** Performance improvement figures have selective presentation. ROUGE itself has a coarse granularity, and the practical value of this magnitude of difference is also questionable.
>
> Thank you. Our experimental setup strictly adheres to established community standards. First, regarding the performance improvement, unlike prior studies that utilized non-instruction-tuned LLMs (e.g., T5-Large/XL) as baselines, we employ SOTA LLMs (e.g., the Qwen3 series), which already possess formidable instruction-following capabilities. Achieving a ~2% improvement over such robust, modern SOTA baselines is highly non-trivial and represents significant practical value.
>
> Second, regarding the granularity of ROUGE, the majority of our selected evaluation tasks involve highly constrained and structured outputs as follows. In these specific scenarios, ROUGE serves as a highly reliable proxy for exact-match accuracy.
>
> Task|Input|Ouput
> -|-|-
> task073-CommonsenseQA|If a person stutters when he experiences anxiety or excitement, he'll have difficult doing what? (A)express information (B)dance (C)library (D)go somewhere (E)study|A
> task1510-relation-extraction|person is a kind of organism|person IsA organism
> task875-emotion-classification|i take a look as i try to get used to the feeling of his touch innocent as it is|joy
>
> Given the structured nature of these outputs, ROUGE precisely and fairly estimates the LLMs' actual performance without suffering from the coarse granularity typical in open-ended text generation.

---

> > ### Author Rebuttal · Reviewer_Tt42 · 2026-04-02
> >
> > The rebuttals have resolved my questions and I will keep the positive score.

---

> > > ### Author Response · Authors · 2026-04-03
> > >
> > > Thanks once again for your valuable comments. We are delighted to hear that your concerns have been resolved, and we are very grateful for your consistently positive evaluation of our work.

---

### Official Review · Reviewer_tNs9 · 2026-03-12

**Soundness:** 3
**Presentation:** 3
**Significance:** 3
**Originality:** 2
**Overall Recommendation:** 4
**Confidence:** 4

**Summary:**

This paper addresses the challenge of cross-task interference in multi-task instruct-tuning for LLMs. The authors empirically demonstrate that existing solutions, such as parameter isolation or MoE, still suffer from significant parameter overlap and conflicting gradients across tasks. To mitigate this, they propose BADIT, which involves two main strategies:(1) Basic Ability Decomposition (BAD): Decomposing pre-trained LLM weights into orthogonal LoRA experts using Singular Value Decomposition (SVD), which are interpreted as "basic abilities". (2) Dynamically Orthogonal Grouping (DOG): An iterative optimization algorithm using spherical K-means and SVD to regroup rank-1 components during training. This ensures that the gradients of different experts remain orthogonal, preserving the distinctiveness of each "basic ability" throughout the optimization process. Experiments on the SuperNI benchmark across 6 LLMs show that BADIT outperforms state-of-the-art methods like GainLoRA and effectively reduces catastrophic forgetting in sequential training.

**Compliance With Llm Reviewing Policy:**

Affirmed.

**Final Justification:**

The author's rebuttal addressed my primary concerns regarding expert interpretability. My final recommendation for this paper remains a Weak Accept.

**Key Questions For Authors:**

1. Can you provide qualitative examples or a visualization of what the "basic abilities" represent? For instance, do certain experts consistently activate for specific task types (e.g., summarization vs. extraction)? This would clarify if the decomposition is truly capturing reusable skills.
2. Since the DOG stage involves SVD and integer optimization on the CPU, how does this scale as the model size (m, n) or the number of experts increases? Would it become a significant bottleneck for 70B+ parameter models?
3. You choose $K=8$ and $r=4$ for experimental setting. Does this optimal configuration change for different model scales? Providing this data would help demonstrate the robustness of the "basic ability" count across scales.

**Limitations:**

yes

**Strengths And Weaknesses:**

**Strengths**
1. The paper tackles a highly relevant problem in LLM fine-tuning, i.e., gradient conflict in multi-task learning. By shifting the focus from task-specific isolation to "basic ability" isolation, it provides a promising new perspective for improving model generalization.
2. The authors provide a formal proof of the orthogonality of their SVD-based initialization and the invariance of their regrouping strategy regarding model output.
3. The experimental results demonstrate the effectiveness of BADIT.

**Weaknesses**
1. While the authors justify the trade-off, the DOG stage introduces approximately 22% additional training time compared to LoRAMoE. The reliance on CPU-intensive integer optimization could be a bottleneck for larger-scale deployments.
2. While the paper uses the "basic abilities" analogy, it does not provide a qualitative analysis of what these "abilities" actually represent (e.g., linguistic patterns, reasoning steps). Understanding the semantic meaning of these decomposed experts would strengthen the contribution.

---

> ### Author Rebuttal · Authors · 2026-03-29
>
> We appreciate your thorough and constructive feedback. In response to the points raised, we provide the following explanations.
>
> > **W1.** The DOG stage introduces approximately 22% additional training time compared to LoRAMoE. The reliance on CPU-intensive integer optimization could be a bottleneck for larger-scale deployments.
>
> Thank you. While BADIT's training time is ~22% higher than LoRAMoE, their real-world deployment and inference time remain identical, due to the completely same model structure and parameter scale. Meanwhile, unlike prior MoE methods that scale expert counts linearly with the number of tasks, our focus on orthogonal basic abilities largely reduces the requirement of the experts. From the empirical performance, given the ~2% multi-task performance boost and the consistent mitigation of the forget rate (all without added inference latency), we believe this training overhead is well-justified.
>
> > **W2.** While the paper uses the "basic abilities" analogy, it does not provide a qualitative analysis of what these "abilities" actually represent. / **Q1.** Can you provide qualitative examples or a visualization of what the "basic abilities" represent?
>
> Thank you. Our central claim is that multi-task instruction-tuning inevitably suffers from parameter interference. Inspired by the co-activation phenomenon in Fig.1, we hypothesize that different tasks inherently share a set of underlying basic abilities. Accordingly, we decouple these abilities and maintain their orthogonality to enhance both learning efficiency and overall performance.
>
> To further explore whether these decomposed LoRAs represent distinct skills, we conduct a toy experiment on SuperNI (K=8, top-4 experts activated) and evaluate expert activation on 20 MMLU tasks using our proposed BADIT. Their activated patterns are as follows:
>
> Task|A|B|C|D|E|F|G|H
> -|-|-|-|-|-|-|-|-
> anatomy|1|0|1|1|0|1|0|0
> clinical_knowledge|1|0|1|1|0|1|0|0
> college_biology|1|0|1|1|0|1|0|0
> college_mathematics|1|0|1|1|0|1|0|0
> college_physics|1|0|1|1|0|1|0|0
> electrical_engineering|1|1|0|1|0|1|0|0
> global_facts|1|0|1|1|0|1|0|0
> high_school_european_history|1|0|0|1|0|1|0|1
> high_school_government_and_politics|1|0|1|0|0|1|1|0
> high_school_mathematics|1|0|1|1|0|1|0|0
> high_school_microeconomics|1|0|0|1|0|1|0|1
> high_school_psychology|0|0|0|1|0|1|1|1
> high_school_statistics|1|0|1|1|0|1|0|0
> high_school_us_history|1|0|0|1|0|1|0|1
> international_law|1|0|1|0|0|1|1|0
> marketing|1|0|0|1|0|1|1|0
> philosophy|1|0|0|1|0|1|0|1
> professional_accounting|1|0|0|1|0|1|1|0
> professional_law|1|0|0|0|0|1|1|1
> sociology|1|0|0|1|0|1|0|1
>
> Our qualitative analysis (where 1=activated, 0=inactive) reveals: Experts 1 and 6 consistently activate across nearly all tasks, suggesting they encode general skills; Experts 1, 3, 4, and 6 concurrently activate for factual and logical knowledge; Expert 8 exhibits a strong preference for history and sociology; Expert 7 is predominantly triggered by governance and law-related tasks. These results suggest that expert activations follow identifiable functional patterns. We will include an exhaustive analysis across additional tasks and LLMs in the Appendix of the revised version. In future work, we plan to explore explicit constraints (e.g., constrained routing) to further enforce the disentanglement of these abilities.
>
> > **Q2.** Since the DOG stage involves SVD and integer optimization on the CPU, how does this scale as the model size (m, n) or the number of experts increases?
>
> Regarding scalability with model size $(m,n)$ and expert count, we first clarify that SVD is performed only once before training; only integer optimization occurs during the DOG stage. Meanwhile, as detailed in **Appendix E.2**, $O$-time complexity scales linearly with $m+n$ and quadratically with the number of experts, while space complexity is completely independent of $m+n$. Because our method decouples basic abilities, it requires significantly fewer experts than traditional MoEs, effectively mitigating the quadratic scaling cost. Additionally, our method can further benefit from existing sparse MoE techniques that bypass inserting modules into all MLP layers, thereby enhancing computational efficiency.
>
> > **Q3.** You choose $K=8$ and $r=4$ for experimental setting. Does this optimal configuration change for different model scales?
>
> Thank you. We have taken the selection of hyperparameters into careful consideration. Specifically, we conduct parameter sensitivity experiments and analyses the number of experts $K$ and the rank $r$, the results of which are presented in **Figure 12 within Appendix H.3**. Our empirical findings indicate a consistent trend across different LLMs: optimal performance is consistently achieved at $K=8$ and $r=4$. Further increasing $K$ and $r$ does not yield additional performance gains but instead exacerbates the computational overhead during training. These observations served as the empirical basis for our final hyperparameter configuration.

---

> > ### Author Rebuttal · Reviewer_tNs9 · 2026-04-03
> >
> > The author's response provided some analysis about "basic abilities". I will keep my positive score.

---

> > > ### Author Response · Authors · 2026-04-03
> > >
> > > Thanks once again for your valuable comments. We are delighted to hear that your concerns have been resolved, and we are very grateful for your positive evaluation of our work.

---

### Decision · Program_Chairs · 2026-04-30

**Decision:**

Accept (regular)

**Comment:**

This paper considers multi‑task instruct‑tuning and proposes BADIT, which decomposes model updates into orthogonal LoRA experts representing shared basic abilities, with a dynamic orthogonality‑preserving training procedure (DOG). Reviewers agree that the problem is important and that the method is technically sound, with empirical advantage over baselines across multiple LLMs and training settings.

The main strengths are the novel perspective of capability‑level isolation (rather than task‑level isolation), solid experimental design (multiple models, mixed and sequential training, multiple seeds), and evidence that BADIT reduces interference and forgetting. Reviewers also appreciated the added qualitative analyses during rebuttal providing further clarifications.
Concerns remain about the conceptual leap from co‑activation to interpretable basic abilities, the limited semantic grounding of these abilities, the modest magnitude of performance gains on strong baselines, and the additional training overhead introduced by DOG. While the rebuttal addresses many of these points, support remains cautious.

Overall, this is a solid and thoughtful contribution with promising ideas, suitable for acceptance, though with moderate enthusiasm. I thus recommend weak accept.